# A modular multi-color fluorescence microscope for simultaneous tracking of cellular activity and behavior

Euphrasie Ramahefarivo [1,2,4], Leonard Böger [1,2,3,4], Takkasila Saichol [1,4], Behzad Shomali [1], Luis Alvarez[1] & Monika Scholz [1]✉

We present a modular epifluorescence tracking microscope which enables ratiometric imaging of muscles, neurons, and other structures in moving animals. The microscope is assembled entirely from commercial parts within 3 h, making the system broadly accessible. Leveraging the improved brightness and bleaching characteristics of recent genetically encoded indicators and fluorophores, the simple microscope is even suitable for calcium imaging of neurons in behaving animals, as we demonstrate in *C. elegans*. We also show how muscle dynamics in *D. melanogaster* larvae can be analyzed and how dual color fluorescence tracking elucidates inter-species interactions by visualizing both predatory nematodes and their prey. Finally, we showcase a configuration for brightfield imaging by tracking tardigrade gait as an example of utility for non-labeled species. The affordability of the hardware and ease of use of the accompanying software make this a suitable tool for education in addition to its use in research.

Access to automated, robust behavioral tracking has transformed research in multiple diverse research areas, such as neuroscience, behavioral ecology, and ethology[1–4]. Combining simple static cameras for high-speed videography with user-friendly deep-learning tools has enabled researchers to rapidly expand their capabilities to study behaviors and scale up the analysis to many recordings[5–8]. However, smaller animals with sizes ranging from sub-mm (hydra, tardigrades) to multi-mm range, such as the nematode *Caenorhabditis elegans*, larval zebrafish, *Drosophila* larvae, or planaria, are typically too small for standard videography and require microscopes for imaging, resulting in small fields of view that do not allow for unrestrained exploration. To this end, tracking microscopes that continually follow an animal as it moves have been employed, keeping it centered in a camera's field of view (FOV). Tracking microscopy can extend the duration and spatial range over which an animal can move within an experiment beyond setups with static cameras. The resulting videos have the same resolution, and the distance from the animal to the

camera can remain constant even if animals move far from the starting location, easing downstream analyses.

Brightfield tracking microscopes for animals such as *C. elegans* have a long history, with the first automated worm trackers published already in the early 2000s designed for automated phenotyping of posture and locomotion[1,3,4,9,10]. Recent setups have improved speed, ease-of-use, and analysis capabilities[4], detecting additional behaviors like egg-laying and feeding[11]. Extending these brightfield trackers to detect fluorescent molecules allows extracting additional information, for example, connecting posture to muscle activity[12], mapping neuronal activity to behavior[13–17] or allowing multi-animal identification[18].

In contrast to brightfield tracking microscopes, fluorescence microscopes are often built for specific tasks. Their design requires custom components or expensive hardware, making them inaccessible to a broader group of researchers. The development of novel fluorophores[19–21] and complementary metal-oxide-semiconductor (CMOS) imaging chips has ushered in a transformative era in

¹Max Planck Research Group Neural Information Flow, Max Planck Institute for Neurobiology of Behavior—Caesar, Bonn, Germany. ²International Max Planck Research School for Brain and Behavior, Bonn, Germany. ³Max Planck Research Group Genetics of Behavior, Max Planck Institute for Neurobiology of Behavior—Caesar, Bonn, Germany. ⁴These authors contributed equally: Euphrasie Ramahefarivo, Leonard Böger, Takkasila Saichol. ✉e-mail: monika.scholz@mpinb.mpg.de

fluorescence microscopy, substantially altering the speed of acquisition and the price for cameras suitable for fluorescence measurements[22]. These developments enable fluorescence detection from samples even with weaker labeling and using lower magnification, resulting in a larger FOV[23,24]. Additionally, the fast read-out speeds of CMOS cameras enable tracking and recording at high-speed fluorescent cellular structures or organs as the animal moves.

Non-experts often face barriers when accessing newly developed methods due to manufacturing complexity or cost restrictions. To reduce costs, some methods use low-cost 3D-printed parts and custom electronics to replace custom-machined elements or moving elements, such as stages[10,25,26]. However, this approach requires some experience in additive manufacturing and can mean a loss of structural integrity and stability. We therefore set out to develop a dual-color epifluorescence tracking microscope purely from commercially available off-the-shelf components.

In this paper, we present a design for a modular epifluorescence tracking microscope suitable for mm-sized animals. Using calcium imaging in muscles and neurons, we demonstrate its capability for behavioral neuroscience. By imaging predatory interactions between two nematode species with a dual-color configuration, we demonstrate the microscope's unique ability to capture complex behavioral dynamics between species in real time. This capability represents a significant advance for studying ecological interactions under naturalistic conditions. In addition, we demonstrate how the same components can be assembled into a simpler single-color variant of the microscope, and we illustrate the capabilities of this setup for posture detection in the tardigrade. The microscope's affordability, versatility, and user-friendly design make it a valuable addition to the toolkit of researchers studying small millimeter-scale organisms, offering versatile capabilities for fluorescence tracking without the need for specialized fabrication.

## Results

### Hardware: Modular design enables multiple imaging modes

We designed a modular epifluorescence tracking microscope for small model animal tracking (Fig. 1A–C). The hardware is modular, supporting single-color, dual-color, and brightfield imaging modes, with magnifications from 1 to 4.2× suitable for macro-imaging of whole animals, tissues within living animals, and single neurons (Fig. 1D). The list of required parts is available in Supplementary Data 1, as well as on the accompanying website. The microscope compensates for animal motion by moving the entire microscope using a motorized stage, while the arena with the animal(s) remains stationary. This design reduces mechanical vibrations of the animal and allows for additional components such as heaters, coolers, or optogenetic tools to be integrated at a fixed position. Keeping the sample fixed relative to the laboratory simplifies the creation of spatially-dependent environments for experiments such as chemotaxis or thermotaxis[4,27,28] and reduces the risk of generating undesired time-dependent gradients of temperature or light that could be induced by moving the experimental arena. Throughout this manuscript, the word "stage" explicitly refers to this motorized platform that moves the camera and illumination components, not the sample.

The dual-color configuration projects two-color images side-by-side onto the camera using a compact image splitter in the W-design[29] (Fig. 1C). For convenience, this feature is matched in the user interface by a corresponding automated color calibration that allows saving the resulting images as registered stacks, which are already processed to have the correct overlap between each channel (Fig. 1D, Supplementary information, and Supplementary Table 1). The two other configurations (brightfield and single color fluorescence) have simplified light paths and primarily use a subset of the components of the dual color microscope, allowing users to switch between different use cases (Supplementary Fig. 1, Supplementary Table 2, Supplementary Data 1).

Magnification can be adapted by exchanging the objective (Fig. 1C, D). The standard configuration with the largest FOV results in a 1× magnification and 2.34 µm/px resolution while allowing a 7.4 mm × 5.0 mm FOV. We present three configurations, with resulting magnifications ranging from 1× to 4.2× (Supplementary Table 2). Other dry objectives already present in the lab can be used, requiring only the appropriate adapters (Fig. 1D; see "Methods" to calculate the resulting magnification). The light sources are commercial LEDs, which can be matched to the desired fluorophore, such as deep IR for behavioral imaging or a general-purpose white LED, which we use throughout due to its broad applicability for many different fluorophores.

The stage is critical for the function of the microscope, as it will determine at which speeds animals can be tracked successfully. To allow easy integration with any computer and operating system, we use a USB 3.0 stage with integrated motor controllers, capable of moving up to 20 mm/s and providing a travel range of 150 mm in both X and Y directions. This range supports imaging arenas up to approximately 15 cm square, suitable for large behavioral plates as demonstrated with our *Drosophila* larvae experiment using large plates (Fig. 2). Users should be aware that partial occlusion from the stage hardware may occur at extreme stage positions. When assembled, the setup has an effective imaging area of $100 \times 140\ mm^2$, which is sufficient for tens of minute-to-hour duration experiments with unrestrained, mm-sized animals. The camera is a Basler sCMOS model capturing up to 57 full frames per second, featuring a USB 3.0 connection for platform-independent integration without frame grabbers. All hardware components are available from commercial vendors and do not require soldering or manufacturing, only assembly. Detailed documentation on assembly and usage is provided online (see "Data and Code availability" section), allowing users to assemble the microscope within 3 h using the illustrated guide (https://scholz-lab.github.io/GlowTracker/).

### Platform-independent software for effective interaction with the hardware

To allow users to interact and perform experiments with the microscope, we have built GlowTracker: an application with a graphical user interface that ties the hardware together. The application can record and display images from the camera, control the stage, and perform automated object tracking (Supplementary Movie 1, Supplementary Movie 2, Supplementary Table 1). GlowTracker is built on top of the Python-based Kivy App framework and has been designed to work on Windows, Linux, and macOS (see "Code availability" section for a link to the software). Key capabilities include an autocalibration where the pixel size and stage orientation are automatically determined, and a color calibration feature, which calculates the optimal overlay of multiple color channels to minimize post-processing of the resulting images. In addition, the software includes a macro language, allowing basic scripts to move in a defined area or take z-stacks. Taken together, these features enable flexible usage of the microscope even for users inexperienced in coding.

To determine the GlowTracker GUI's performance, we ran benchmark tests on image analysis and tracking. We found that tracking performance was most influenced by the exposure time, as this limits the framerate at which images are incoming. In practice, we could obtain image acquisition frame rates of 16 Hz up to 50 Hz for bright samples, which was only limited by the required exposure times (Supplementary Fig. 2). To avoid calculating the displacement on blurred frames, we ignore frames where the stage moved in tracking calculations. In addition, the built-in stage controllers have a communication latency of about 25 ms, which is comparable to typical exposure times (Supplementary Fig. 2). If faster tracking is desired, users could upgrade to standalone controllers that have much lower communication latencies. However, we found that for all of our applications, communication speed was not a limiting factor for

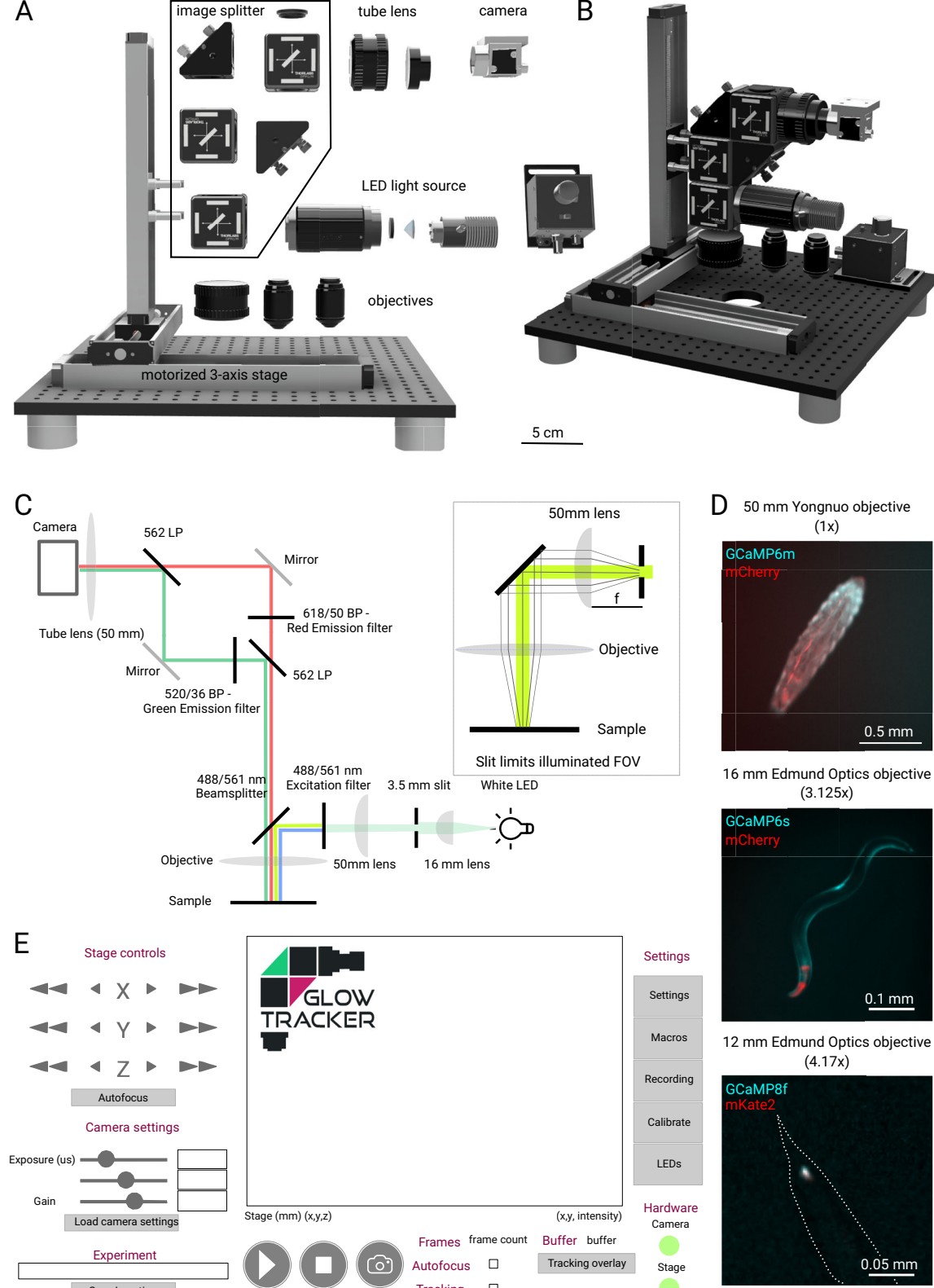

**Fig. 1 | Design of a modular epifluorescence tracking microscope.**
**A** Components and **B** 3D rendering of the dual-color epi-fluorescence tracking microscope. **C** Lightpath of the dual color tracking microscope using a slit imaged onto the sample plane to restrict illumination (inset) and a compact image splitter design. **D** Example images of dual color imaging at different magnifications of a *Drosophila melanogaster* 3rd instar larva with muscles expressing mCherry and GCaMP6m at 1× magnification; Image of an adult *C. elegans* with pharyngeal (red)

and body wall muscles (cyan) labeled with mCherry and GCaMP6s at 3.125× magnification and *C. elegans* touch receptor neuron PLM labeled with mKate2 (red) and GCaMP8f (cyan) imaged at a magnification of 4.17×. The outline shows the tail of the animal (dashed). Magnification was adjusted by using objectives with a focal length of 50, 16, and 12 mm, respectively. The same 50 mm tube lens was used for all recordings. **E** Schematic of the GlowTracker GUI showing the interface and different features such as calibration and autofocus.

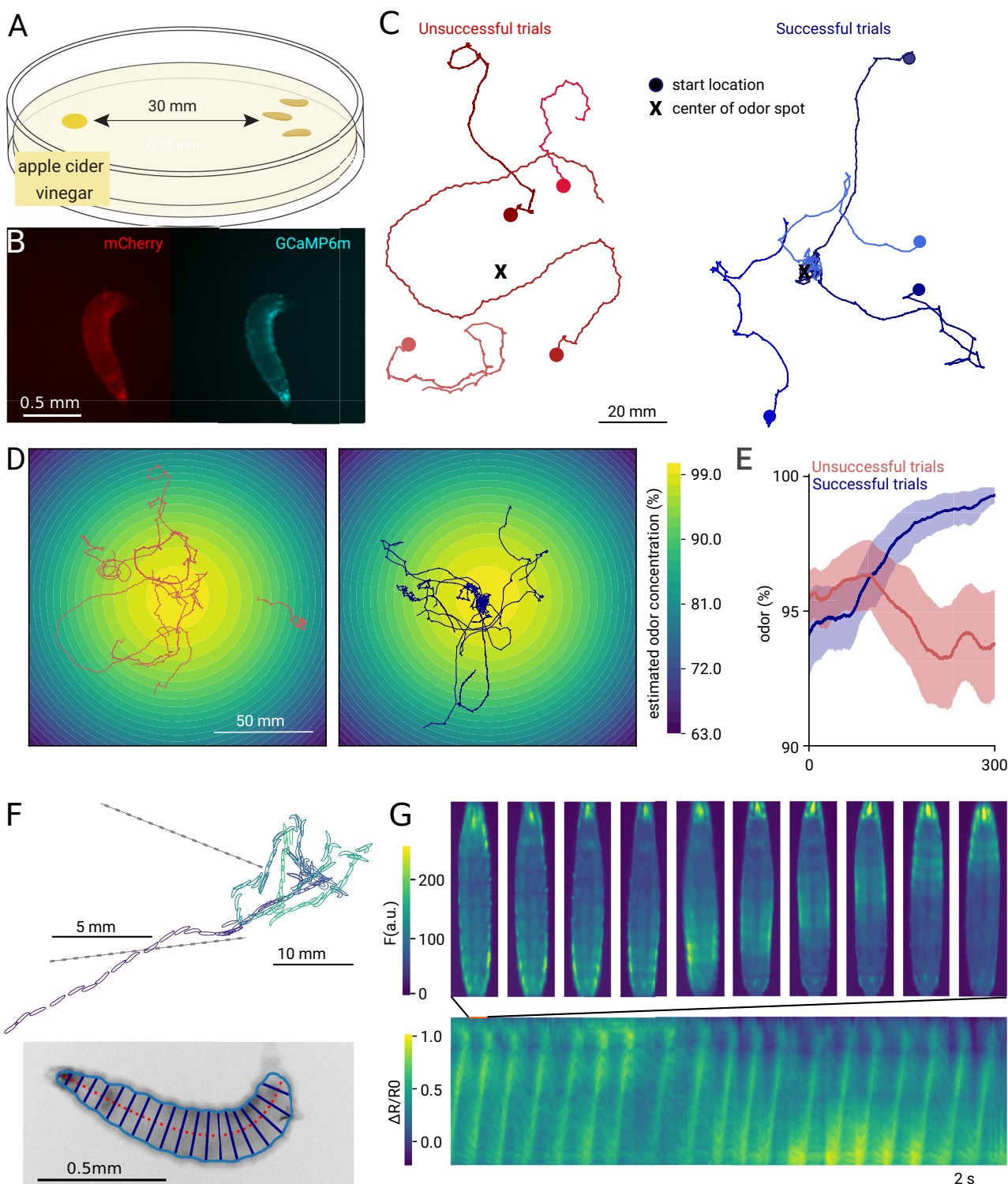

**Fig. 2 | Tracking behavior of *D. melanogaster* larvae while navigating an odor gradient. A** Schematic of the odortaxis assay. **B** Example images of the dual-view ratiometric calcium imaging during the experiment. **C** Representative example tracks for both successful (blue) and unsuccessful (red) trials. Trials were counted as successful if the larvae entered and remained within 5 mm of the center of the target spot within the 5 min of recording time ($N = 4$ selected traces). **D** Estimated odor gradient from a diffusion model of vinegar, and **E** odors experienced by the successful (blue) and unsuccessful (red) larvae over time. **F** Example track of a successful larva with the posture overlaid. Posture was extracted from images as shown in the inset below. Dashed lines indicate a track region shown at higher magnification. **G** Kymograph of muscle GCaMP activity for the animal in (**F**) during straight navigation. The Ca²⁺ waves along the animal body during muscle contraction can be observed. Exemplary images during a single stride are shown at the top.

tracking performance. Depending on the camera frame rate, the system can track (i.e., update the stage position) at 6.5 Hz to 11 Hz. The full description of benchmarking and hardware requirements is discussed in the Supplementary Information.

In summary, we designed a modular microscope suitable for imaging samples in the few-mm to μm range that allows single- or multi-color imaging. To test the applicability of the setup, we present multiple typical use cases: ratiometric calcium imaging of neurons and muscles of unrestrained animals, tracking of non-labeled model species, as well as visualization of inter-species predatory interactions.

## Simultaneous tracking of posture, muscle activity, and locomotion in *D. melanogaster* larvae

With a speed of about 1 mm/s, *Drosophila* larvae can explore arenas of multiple tens of cm within minutes, requiring a tracking microscope to bridge the scale between muscle contraction and large-range locomotion[30]. Fly larvae range between 1.4 mm at their first larval stage up to 3–5 mm at their third larval stage[31], making them suitable for imaging with the 1x microscope (Fig. 2A, B). Larvae are effective at finding attractive odor sources, using a navigational algorithm that includes active sampling[32]. As larvae move by peristaltic traveling waves along their anterior-posterior axis, interspersed with turns, the navigation is well-described by specific muscle contraction patterns[33–35]. For example, while traveling straight requires symmetric left-right muscle contractions, turns and bends are created by asymmetric contractions. We aimed to connect the biomechanics of the body to the navigational strategy by simultaneously imaging the larvae's navigation along a vinegar gradient and monitoring muscle activity with a genetically encoded calcium indicator (Fig. 2A, B, Supplementary Fig. 3, Supplementary Movie 2, Supplementary Movie 3).

Notably, our dual-color configuration achieves simultaneous imaging of two distinct spectral channels by restricting excitation illumination to a single narrow slit, which is optically imaged on the sample (Fig. 1C). This slit confines the excitation light to a localized area tightly centered on the tracked larva, ensuring stable and uniform illumination as the animal moves within the arena. By spatially limiting illumination and continuously tracking the animal to keep it within the slit, we prevent illumination gradients across the behavioral arena that could bias behavior.

Using this setup, we studied larval navigation and found that some larvae do not reach the odor spot within the allotted time, despite having sufficient speed to be able to reach the spot (Fig. 2C–E). We confirm that animal size correlates with speed, as had been reported previously[36] (Supplementary Fig. 4); however, speed did not differ significantly between successful and unsuccessful larvae (Supplementary Fig. 4B). To investigate potential differences in navigational strategies between successful and unsuccessful larvae at the biomechanical level, we analyzed their posture and calculated a virtually straightened animal. Using the two-color imaging, we can ratiometrically correct the GCaMP images and extract features like fluorescence intensity along the anterior-posterior axis (Fig. 2F, G), showing characteristic peristaltic waves used for propagation. Similar analyses have enabled the prediction of larval locomotion from muscle activity[12], but the amount of motion was constrained by the FOV. Allowing imaging of unrestrained animals in large arenas will be useful for a variety of applications: From basic biomechanics research into how posture translates into movement, to studies of movement disorders. While fictive locomotion assays and assays using non-tracking microscopes have already provided insight[37,38], experiments on the microscope can be done in structured environments, such as gradients, and allow for many more strides per larva. In summary, we show that the modular tracking microscope is suitable for tracking larval *Drosophila* across cm-scaled environments while extracting relevant physiological data.

## Ratiometric imaging of single neurons in moving animals

To demonstrate the dual-color capability in a smaller sample under challenging conditions, we aimed to image single neurons in unrestrained animals. Imaging single neurons in freely moving animals is technically demanding, due to their small size, 3D movement, and motion artifacts. Such experiments typically require costly, high-magnification, custom-built microscopes or custom post-hoc tracking[39–42]. Remarkably, our low-cost modular system - constructed entirely from commercial components and configured for 4.1x magnification successfully captures these changing dynamics. To demonstrate the capabilities, we chose to reproduce findings from a very well-characterized circuit in *C. elegans*, the circuit responding to gentle touch[43].

Recent advances in single-neuron labeling, such as the use of Gal4 driver lines[44] and newly developed sensory-specific strains[45], have allowed for more precise labeling. In addition, fast and bright calcium indicators are now amenable to low-magnification imaging with our microscope. We recently adapted the new generation calcium indicator GCaMP8f for use in nematodes and demonstrated its properties in restrained animals[19]. Using this indicator, we chose to image the dynamics of *C. elegans* touch receptor neurons (TRNs), which induce an escape response when stimulated (Fig. 3A, B). As immobilization has been shown to alter neural activity[14,37], we focused on capturing neural responses in a natural, unrestrained environment. Prior work used either custom microscopes[11,46], or low-resolution large FOV imaging methods in small arenas. Our system bridges this gap by enabling targeted, high-frame-rate imaging of specific neurons like TRN.

We employed ratiometric imaging techniques to reduce motion artifacts and correct for focusing issues, while enabling high-frame-rate imaging of TRNs. In our experiment, we stimulated *C. elegans* worms with a mechanical vibration for 1 s, resulting in the activation of all TRNs following the same protocol as in ref. 19. We tracked the PLM neuron, a TRN located in the *C. elegans* tail, proposed to promote forward escape responses[43,47]. Expectedly, the animals showed increased speed and a significant rise in the PLM neuron's calcium signal $\Delta R/R0$ (Fig. 3C, D). Interestingly, when comparing worms that showed reversals within 2 s after stimulus onset to those that did not, we could observe distinct calcium traces (Fig. 3E). Worms showing a reversal escape response had a decreased peak activity in the PLM neurons and decayed faster compared to animals who performed a forward escape (Fig. 3F). The microscope is therefore suitable for imaging challenging, small neural samples in moving animals while allowing for robust tracking and motion correction at a cost that is substantially lower than comparable published systems.

## Simultaneous imaging of animals and the environment

To study interspecies interactions, we employed the dual-color microscope to visualize predatory behavior between two nematode species: the predator *Pristionchus pacificus* and its prey, larval *Caenorhabditis elegans*[48,49]. *P. pacificus* can prey on non-kin larvae of other nematodes, exhibiting contact-mediated kin detection[50,51]. We tracked *P. pacificus* expressing red fluorescent protein (RFP) in its pharynx[49] and detected prey contact using a *C. elegans* strain expressing GFP in its body wall (Fig. 4A, B). Our dual-color system maintains clear animal identity, as predator and prey appear in distinct colors projected side-by-side onto the camera sensor. It is possible to customize tracking by selecting which fluorescent label to follow, such as GFP or mCherry, ensuring flexible and accurate tracking even during close interactions. This approach provides previously inaccessible insight into the dynamic behaviors and ecological interactions of these species.

In a previous study[49], we developed a machine-learning model to identify behavioral states of the predator using only the images of its pharynx. In brief, by using a previously developed image analysis tool[24], we extract behavioral features such as speed, pumping rate, posture, and head swings. We are then able to assign stereotypical combinations of these features to distinct behavioral states, some of

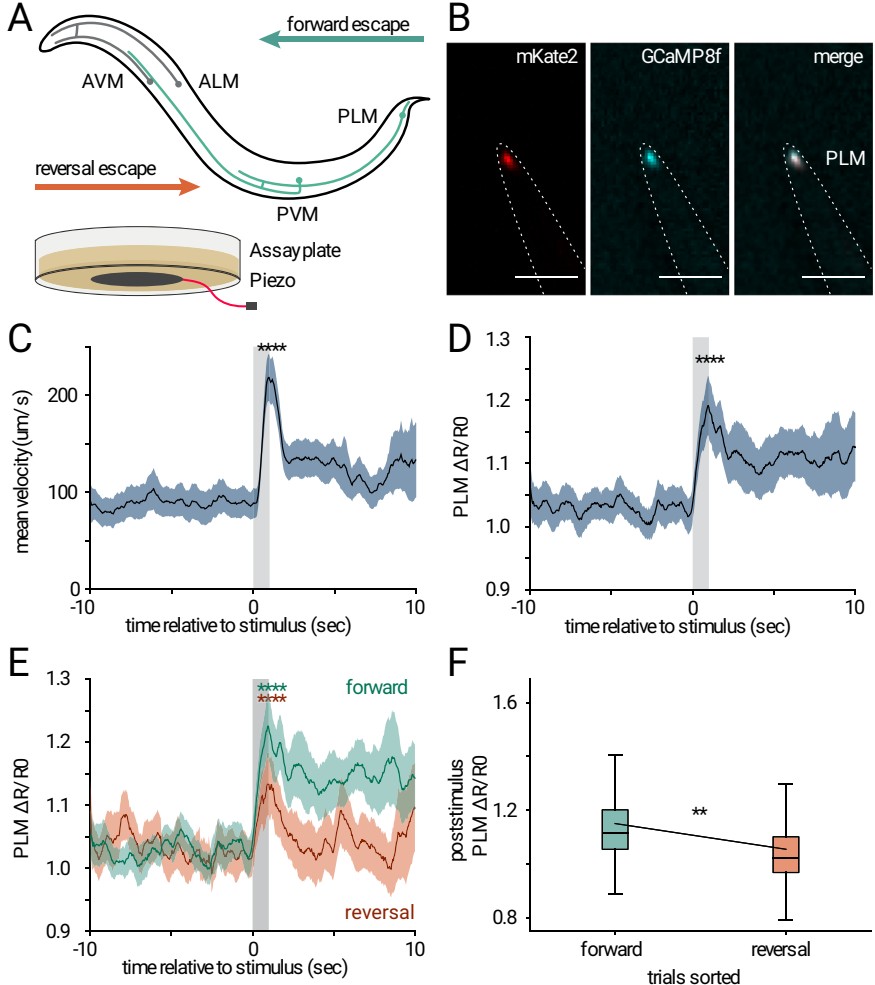

**Fig. 3 | Ratiometric imaging of touch-receptor neurons in unrestrained animals. A** Schematic of touch receptor neurons in *C. elegans* and behavioral response to gentle mechanical stimulus, and an assay plate with piezo. **B** Touch receptor neuron PLM labeled with GCaMP8f and mKate2 (cyan, red, respectively). The dashed line shows the worm's outline. Scale bar corresponds to 50 μm. **C** Speed response (mean ± s.e.m.) to light touch stimulation (gray bar). Statistics performed with two-tailed paired *t*-test, ****$p$ = 1.8e-14 (**D**) Neuronal response (mean ± s.e.m.) to light touch stimulation (gray bar). Statistics as in (**C**), ****$p$ = 2.5e-11. **E** Neuronal response to light touch stimuli (gray bar) as in D, for trials with reversal (orange) ****$p$ = 3.1e-04, and forward response (green) **** $p$ = 2.8e-08 within 2 s after

stimulus onset. Statistics as in (**C**). **F** Boxplot showing mean poststimulus GCaMP8f fluorescence signal, color as in (**E**). For forward responding worms, $N$ = 15 worms, $n$ = 41 traces, with 1–5 repetitions per animal. For reversal responding worms, $N$ = 12 worms, $n$ = 20 traces, with 1–3 repetitions per animal. The box spans the first to third quartiles, with a line marking the median. Error bars extend to the furthest point within 1.5 times the interquartile range. Outliers are excluded. A mixed linear regression analysis with a *z*-test used for significance testing revealed that post-stimulus GCaMP8f fluorescence signal was significantly influenced by reversals (**$p$ = 0.0032).

which are uniquely present in predating animals. Previously, we could only infer the presence of prey, as the prey was not visible in our imaging. However, with the dual-color images, we are now able to confirm the model predictions by visualizing prey contact directly.

A section of the recording is displayed in Fig. 4B–D, showing the predator's centerline and corresponding metrics: velocity and pumping rate. Tracking *P. pacificus* on prey revealed that during biting and feeding states, velocity decreased while pumping rates increased; in contrast, exploration featured higher velocity and lower pumping rates. By aligning multiple ethograms to biting events (Fig. 4E), we could show that the signal from the GFP expressed in the prey body wall muscles significantly increased at bite onset (Fig. 4E, F), reflecting predator-prey contact and ingestion of the green prey. These measurements allowed us to verify the model predictions and confirm that biting states occurred in the presence and close contact with prey. The microscope is therefore suitable for imaging small, moving animals within their conspecifics or other relevant species, thus enhancing our ability to understand multi-species interactions.

## Long-term tracking of organ activity in behaving animals using the single-color microscope

Thanks to its modular design, the microscope also supports single-color imaging with a simplified optical path (Supplementary Fig. 1), offering twice the FOV for a single fluorescent label or dye. The single-color microscope uses a subset of components from the dual-color microscope, as it does not require the image splitter (Fig. 5A, B). Similar to the dual-color setup, the magnification can be adjusted to suit specific applications. For instance, we used it here to image the feeding organ of *C. elegans* (Fig. 5C). With the same objectives and magnifications as the dual-color microscope, the design uses relatively low effective magnification and low NA, resulting in a large depth of field. This feature is particularly advantageous as it minimizes motion artifacts and is ideal for imaging entire animals or tracking features within freely moving animals, such as cells or organs that may move out of the focal plane.

To evaluate whether fluorescence levels and tracking stability are suitable for hour-long experiments, we tracked single *C. elegans*

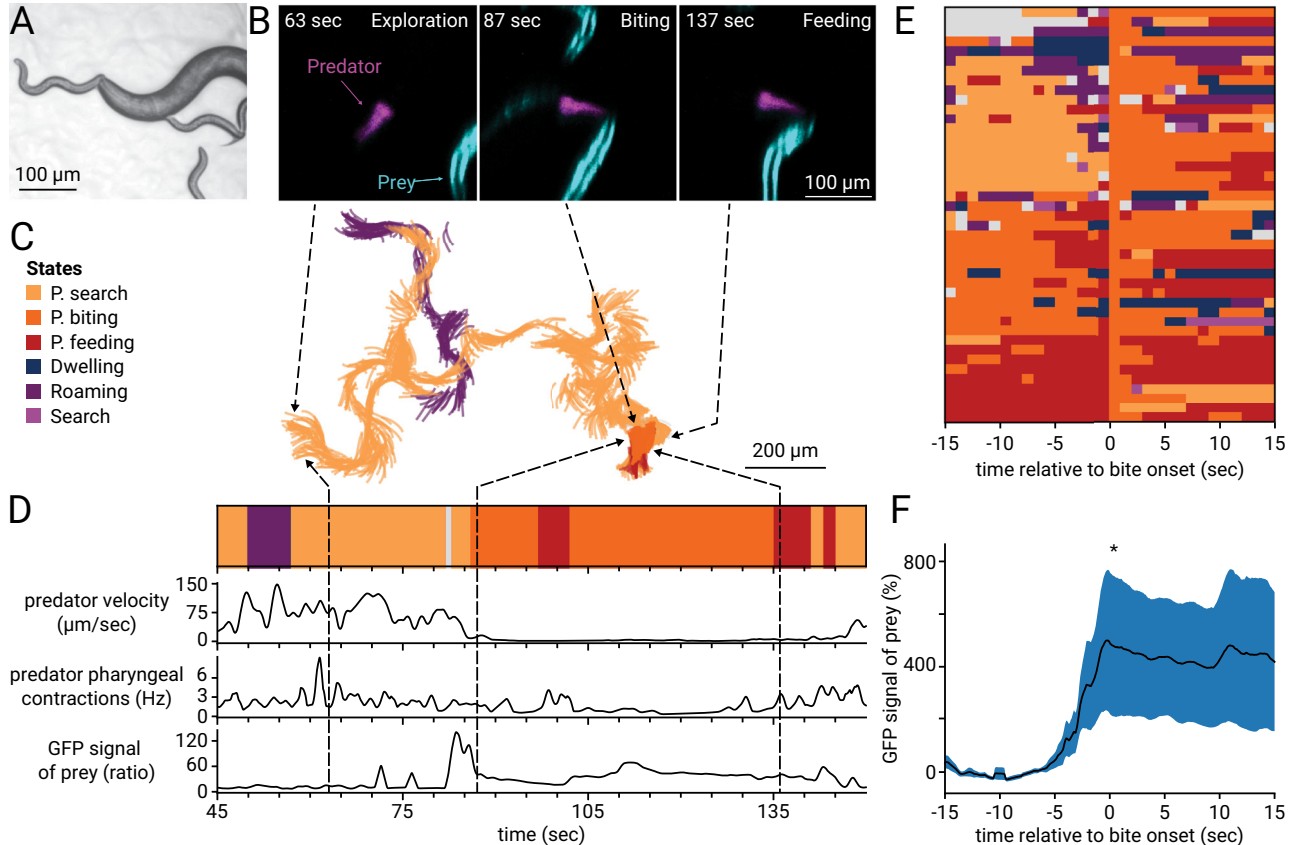

**Fig. 4 | Visualizing animal and environment interaction. A** Example brightfield image showing predator prey contact between *P. pacificus* predator and *C. elegans* prey. **B** Example images of sequential stages of a predatory event involving the predator *P. pacificus* expressing RFP in the pharynx (magenta) and *C. elegans* prey expressing GFP in the body wall muscles (cyan) at 3.1× magnification. The scale bar corresponds to 100 μm. Arrows point to the corresponding track in (**C**). **C** Example tracks of the centerline of the predator between 45 s and 150 s of recording. Colors indicate predicted behavioral states using the model published in ref. 49. The scale bar corresponds to 200 μm. **D** Ethogram showing the predicted behavioral states over time, with corresponding velocity (μm/sec), pharyngeal contractions of the predator (Hz), and prey signal. Prey signal quantifies GFP fluorescence from *C. elegans* prey expressed in the body wall muscles detected in front of the predator's mouth opening. Signal increases indicate prey contact during biting and feeding. Arrows point to the corresponding track in (**C**). **E** Stacked ethograms of $n = 43$ tracks, $N = 8$ predators aligned at the onset of predatory biting at $t = 0$. **F** Prey signal (mean ± s.e.m. in %) of the ethograms shown in (**E**), excluding those with the mode of predatory biting or predatory feeding before bite onset, $n = 17$, $N = 7$. Prey signal was detected as in (**D**), but normalized to the signal within −15 to −5 s before bite onset at $t = 0$. Statistics performed with an upper-tailed paired *t*-test comparing time ranges −15 to −5 and 0 to 15 s, *$p = 0.049$.

nematodes while they foraged. We use the microscope to automatically follow the nematodes with the pharynx expressing a genetically encoded calcium indicator[19]. We find that over the course of 1 h recordings, the mean intensity is only reduced by ~25%, indicating low photobleaching (Fig. 5D). The resulting signals from the calcium indicator are still sufficient to extract peaks, corresponding to contractions of the feeding muscle after ~1 h of recording (Fig. 5E). The tracking stage can automatically retain the animals in the FOV while they travel over multiple centimeters (Fig. 5F), allowing us to observe unrestrained behaviors. This configuration allows simultaneous measurement of multiple behaviors, such as feeding, reversals, and locomotion speed (Fig. 5G). During long-term experiments, the animals remained viable with no significant changes in velocity or pumping rate observed, suggesting that the experimental conditions did not adversely affect the viability of the animals over time. This stability in behavior enables investigations into internal states, development, or sleep[52].

## Brightfield tracking of tardigrades

Brightfield imaging is versatile and amenable for species in which the ability to use transgenics for labeling body parts is not established. To demonstrate the capability of the microscope for brightfield tracking, we recorded locomotion behavior in an emerging model species[53],

namely the tardigrade *Hypsibius exemplaris* (Fig. 6A). By training the pose estimation classifier DeepLabCut, we could predict the position of all eight legs, eyes, as well as the most anterior and posterior ends of the body (Fig. 6B, C). The model performed well, with a mean prediction error of 3.12 μm (5.29 pixels) on the test set and 1 pixel (0.57 μm) on the training set.

We were able to record tardigrade locomotion over 10 min for 10 animals, generating a dataset of more than 1 ½ hours of locomotion data. Prior attempts typically imaged for tens of strides or a few minutes[54,55]. Such a longer dataset could be valuable to discover navigational strategies and their manifestation in locomotion. As it is difficult to annotate such a great number of strides manually, we developed an automatic swing detection algorithm that uses the anterior-posterior leg position to identify swings. To estimate the correctness of swing identification, we compared the automatic swing detection with a manually labeled subset. The automatic annotation has a frame-to-frame accuracy of 81% and a precision of 70% compared to the manual annotation. The difference between automatic and manual swings on- and offsets stays within −3 to +2 frames (10 and 90 percentiles) (Fig. 6D). Moreover, the duration between automatic and manual swings is similar, as can be seen by the positive correlation coefficient (Pearson $r = 0.96$) between automatic swing on- and offset relative to manual swings. Taken together, this indicates that similar

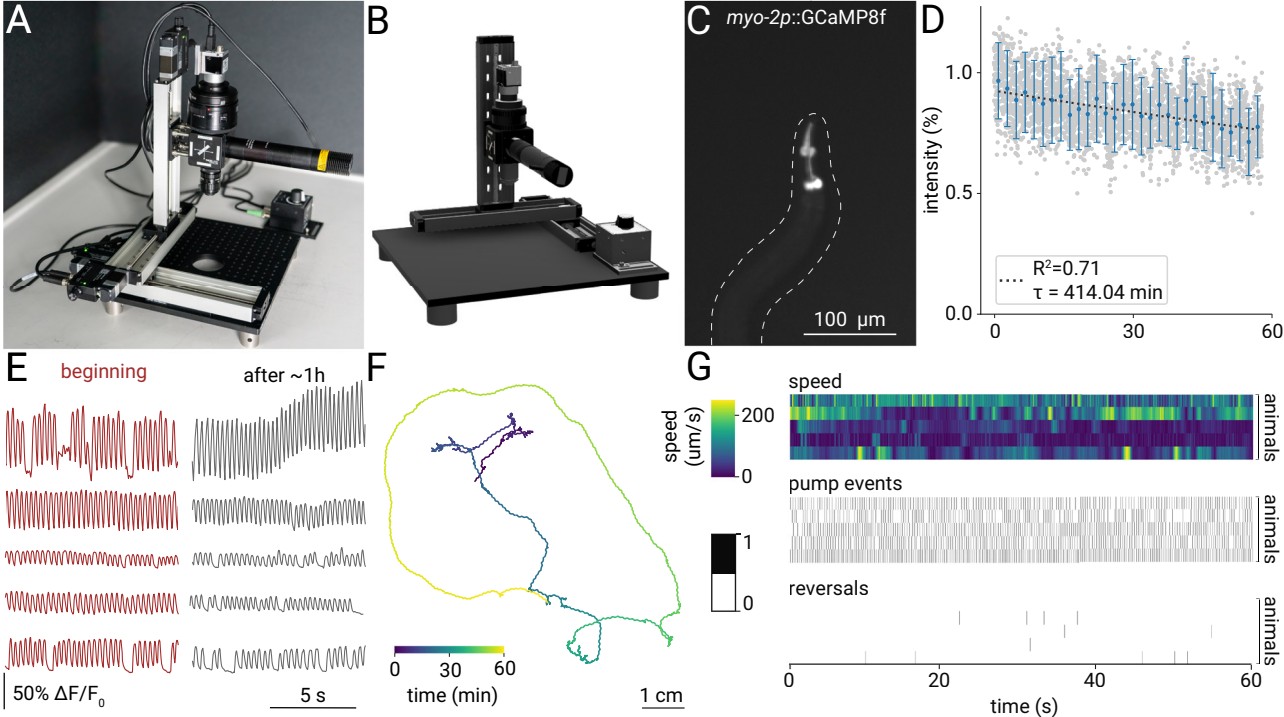

**Fig. 5 | Modular single-color design enables long-term tracking. A** Setup adapted for single color imaging. This version uses a subset of parts from the dual-color microscope. **B** 3D rendered model of the setup. **C** Example image of a *C. elegans* nematode with the pharyngeal muscle expressing GCaMP8f. **D** Photobleaching curves (mean ± s.e.m.) for 1 h of continuous tracking and illumination at 2.78× (*N* = 5 animals). **E** Example signals from muscle contractions measured from the GCaMP8f indicator at the beginning and end of the 1 h track for *N* = 5 animals (top to bottom). **F** Example 1 h track of a worm. **G** Ethograms for *N* = 5 animals for 1 min of data showing speed, feeding, and reversals.

locomotion patterns are extracted, even though the precise onset and offset of these swings may differ by ± 3 frames (0.2 s) compared to a human annotation. Moreover, we find that the duty factor is in a similar range as previously reported, with the mean duty factor for all legs close to 0.6. Moreover, the correlation between duty factor and walking speed is weak, ranging from r = −0.32 to r = −0.25 for the lateral legs, and the weakest for the hind legs (*r* = −0.1) (Supplementary Fig. 5G). This is in line with ref. 54, but is considerably weaker than what was reported in ref. 55.

The tracks reveal a great variation of locomotor patterns that might be linked to differences in navigational strategies (Fig. 6E). While some tardigrades show highly curved tracks and only cover a few millimeters in distance, others cover more than 1 centimeter. Thereby, the strides of the tardigrades appear mostly regular, but seem to transition smoothly between different gait types, as has been established before[54,55]. An example gait of 1 min length is shown in Fig. 6F. To investigate the underlying gait types, we employ a data-driven approach where we measure the pairwise similarity between the normalized anterior-posterior position of all four leg pairs between individual gait cycles (Supplementary Fig. 5A). The similarity was then used to embed the cycles into a two-dimensional embedding and perform clustering using a dimensionality reduction technique (UMAP; Supplementary Fig. 5B), to identify cycles that can be grouped into one gait type. The silhouette score for the reported clusters is 0.50. Because UMAP is dependent on random initialization, we tested whether the clusters reported here are robust to these initializations and reliably appear in our embeddings. The mean agreement of cluster assignment between different UMAP embeddings is 75%. The normalized mean anterior-posterior position of the different cycles shows similarities with gaits previously described in tardigrades (Fig. 6G)[54,55]. Cluster 1 and cluster 2 appear similar to the tetrapod-like gait. In the tetrapod gait, only two legs are simultaneously in the swing phase, and a wave of

stance movements propagates from posterior to anterior legs on both sides[56]. The gait type recovered in cluster 0, is similar to the tetrapod gallop[55], the two leg pairs 2 and 3 are sequentially moved, with the left and right pairs moving simultaneously (Fig. 6G). In our data, we could not recover a tripod gait, with 3 of the 6 most anterior legs in swing. However, it was previously found that tardigrades prefer tetrapod-like gaits[54,55]. We could not identify any previously described gaits that look similar to the gait visible in cluster 3. This gait was also used the most during turns (Supplementary Fig. 5C). Here, turns are defined as sudden directional changes of the tardigrade. Cluster 2 shows the highest mean velocity and the lowest mean angle, indicating this gait may be preferentially used when the tardigrades walk in a straight path (Fig. 6H). Interestingly, we would recover similar gait types when considering only the 3 most anterior leg pairs (Supplementary Fig. 5D–F), as done in previous studies[54,55].

By imaging animals navigating over large distances, we capture more strides per animal than previous studies, likely enabling the observation of gait transitions in unrestrained animals walking over larger distances and potentially performing decision-making or navigational tasks. For broader use cases in gait analysis, future studies could benefit from additional improvements to the experimental setup. For example, an inverted imaging setup, to focus the camera on the legs, could improve the precision of the pose estimation further. As can be seen in Fig. 6F, a few instances of swings have been missed by the automated swing calling, possibly due to noise in the pose estimation. Further studies could investigate how gait choice depends on environmental parameters such as substrate stiffness, or how it supports different navigational goals like climbing[57]. Administering pharmacological drugs such as exogenous bioamines (serotonin or dopamine) could inform how the usage of these neuromodulators has been adapted or conserved[11,49,58], and once transgenics are available in these animals, neuronal or muscle activity could be measured using

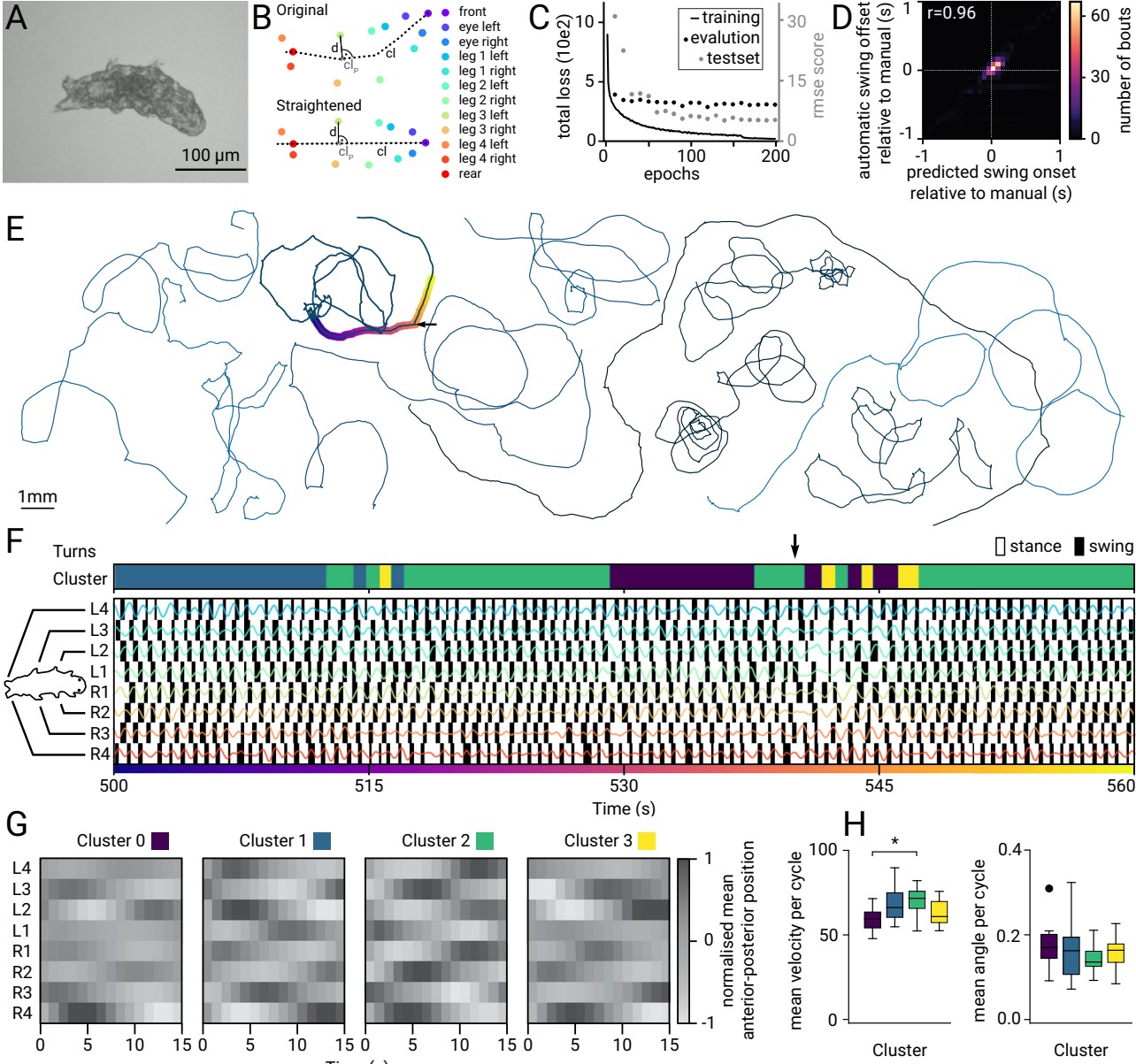

**Fig. 6 | Brightfield tracking of tardigrades. A** Example image of a tardigrade on the microscope. **B** Labels used for pose estimation with DeepLabCut and a schematic showing the straightened coordinates, with **D** being the distance of leg 3 left to the closest point (**cl$_p$**) on the centerline (**cl**). **C** DeepLabCut total loss across 200 training epochs for the trainset, validation set and the RMSE score in pixels for the testset. **D** 2 d histogram of automatic swing on-offset relative to manual on-offset ($N = 10$, $n = 545$, upper-tailed Pearson's mean-$r$ = 0.96 averaged via Fisher-z method, $p$ value = 4e-291, combined with Fisher method with chi$^2$ = 1383). **E** Tracks of tardigrades, highlighted track corresponds to example shown in (**F**). **F** Example

Hildebrand-style gait diagrams, with superimposed anterior-posterior leg position (colored lines), associated cluster assignment, and turn event (indicated by arrow). **G** Normalized mean anterior-posterior position of all leg pairs for all four clusters. **H** Boxplots showing cluster distribution over mean speed and mean angle per animal. Box plots follow Tukey's rule with the box from first to third quartiles, and a line at the median. The whiskers denote 1.5× interquartile range. Statistical comparisons were made using a one-way ANOVA with Tukey's multiple comparison test, $N = 10$, *$p = 0.042$.

the fluorescence configuration of the microscope. Taken together, these examples illustrate the broad spectrum of experiments enabled by this modular microscope.

## Discussion

In this study, we presented a tracking epifluorescence microscope built entirely from commercially available parts, paired with an integrated and fully automated software application. Currently, similar microscopes are built in laboratories by experts in optical design and hardware control, or expensive commercial solutions must be used, which are often less adaptable to diverse experiments. For example,

only specialized hardware solutions for ratiometric neuronal imaging in freely moving animals exist for *C. elegans*[27,59–62], *Drosophila* adults and larvae[63–65], and also zebrafish[13,66].

Previous fluorescence tracking systems for freely moving animals have generally relied on expensive commercial microscope bodies (Zeiss, Nikon, Olympus), specialized cameras (Andor, Hamamatsu EM-CCD), and motorized stages (Marzhauser, Ludl BioPrecision, Applied Scientific[27,42,59,60,62,67]. Among these, only one system provides a comprehensive build guide[27,42,59,60,62,67], and most require proprietary software such as LabVIEW, MetaVue, or Metamorph. Notably, only one design[42] offers fully open-source software, while the commercial

software components for these systems can range into the thousands of euros.

To address these limitations, we designed our system to integrate multiple imaging modalities into a single, accessible platform. Priced at approximately €14,500, with complete, step-by-step build instructions, fully commented open-source software, and assembly time of 3 h, our system fills a critical gap in affordable, reproducible microscopy. Users can switch between brightfield, single-color, or dual-color fluorescence imaging - covering a broad range of applications such as functional calcium imaging to behavioral analysis and interspecies interactions - all using modular components that do not require specialized fabrication. Our goal was to create a microscope accessible to users with less experience in building optical setups, yet sufficient for producing research-quality data.

A key scientific advance is our demonstration of dual-color imaging of predator-prey interactions in freely moving animals. By tracking *Pristionchus pacificus* (tagged with a red fluorophore) among *Caenorhabditis elegans* prey (tagged with a green fluorophore), we achieved real-time visualization of the complete predatory sequence— from exploration through biting and feeding - allowing successful detection of prey contact through GFP fluorescence. This capability allows researchers to visualize and quantify dynamic interspecies behaviors to reveal the precise timing and success rate of predatory events.

Making such measurements accessible to the broader research community guided our hardware design. While some parts could be replaced by hardware made in a mechanical or electrical workshop, we purposely decided to suggest only commercially available components to make the design accessible to novice users or users without access to such facilities. Our website suggests which parts could be replaced by machining or laser cutting of components, depending on the specific use case.

We purposely selected lower magnifications than commonly used for calcium imaging in microscale animals, as it allowed for a wider range of applications and organisms. Specialized hardware, for example, the 'WormsPy' system, excels in high-resolution imaging of cellular processes[68], while our setup offers flexibility and accessibility for diverse experimental setups and appropriate fields-of-view for larger animals. Both systems address different needs in the field, with WormsPy ideal for detailed cellular studies and our microscope suitable for broader behavioral and ecological investigations. Moreover, our setup could be adapted by using a tube lens with a longer focal distance to achieve higher magnification, suitable for neuronal imaging of smaller cells.

The current microscope design can be readily adapted for experiments using optogenetic tools. The illumination port can be extended to accommodate an additional filter cube, combining fluorescence excitation light with a dedicated light source for stimulation of the optogenetic tool. Here, it is important to consider the tool's power-density requirements, as these are often high. For this reason, it is recommended to use aspheric condenser lenses with high NA to collimate the stimulus light before combining it with the fluorescence excitation light. Such a concept has not been explored in this study, but there are no limitations that would hinder such an extension in the current proposed design of our microscope.

Our microscope is versatile, allowing tracking and measurement of functional properties of muscles and neurons, as demonstrated in larval *Drosophila* crawling, *C. elegans* locomotion, and touch receptor neuron activity. The dual-color feature unlocks possibilities for studying multi-animal and multi-species interactions, bridging the gap between ethology and ecology. For instance, it could shed light on intriguing behaviors like stress-induced cannibalism in *Drosophila* larvae[69]. Similarly, the microscope could be used to track individuals within collectives, by labeling a single or a few animals, relevant to investigate collective behaviors[70,71]. Thus, we expect that our proposed microscope fills a niche between more specialized and expensive setups and simpler brightfield behavioral trackers, offering a valuable tool for many users, bridging cellular and behavioral neuroscience.

## Methods

### Ethics approval
All animals used in this study were invertebrates and not subject to animal protocols.

### Statistics and reproducibility
All imaging experiments were repeated as stated in the relevant captions of the figures. All micrographs shown in Fig. 1 correspond to data shown in Figs. 2, 4, 5, and were repeated at least 5 times for independent animals.

### *C. elegans* maintenance
All nematode strains were cultured on Nematode Growth Media (NGM) plates with OP50 at 20 °C (see Table 1).

### Imaging plates
All experiments were performed on imaging NGM plates unless otherwise stated. In contrast to NGM plates, imaging plates contain no cholesterol and 17 g of agarose instead of agar for 1 L medium.

**Recordings.** For all experiments, the GlowTracker software (v.0.11) was used to record images and stage position during tracking.

**Long-term tracking.** *Assay* Long-term assays were performed on 10 cm imaging NGM plates containing a concentrated patch of fresh OP50 (10X concentrated from an overnight culture). *C. elegans* young adults were selected and starved for 2 h before being transferred individually to the NGM plate. Following a 15-min acclimation period under blue light, the worm locomotion was recorded for 1 h at 3.125× magnification using the single-color configuration with a 15 ms exposure and a sampling rate of 30 frames/s.

## Table 1 | Strains used in this study

| Species | Designation | Identifiers | Source or reference |
|---|---|---|---|
| *C. elegans* | *zcIs14 [myo-3::GFP(mit)]* | SJ4103 | CGC |
| *C. elegans* | *wpIs98 [itr-1pB::Chrimson::SL2::mCherry + odr-1p::RFP] I. wpIs103 [myo-3p::GCaMP6 + myo-2p::mCherry].* | XE1995 | CGC |
| *C. elegans* | *nonEx106[myo-2p::GCaMP8f]* | INF418 | 19 |
| *D. melanogaster* | *MEF2-Gal4, UAS-mGerry* | MEF2-Gal4 | Heckscher Lab, University of Chicago |
| *P. pacificus* | *Ppa-myo-2p::RFP* | JWL27 | 49 |
| *C. elegans* | *lite-1(ce314) gur-3(ok2245) X; nonEx133[Pmec-17_GCaMP8f_SL2_mKate2_let-858_3'UTR + Punc-122::tagBFP]* | INF204 | 19 |
| *H. exemplaris* | | | Gift from Jean-Paul Concordet, Muséum National d'Histoire Naturelle, France |

**Analysis of long-term *C. elegans* foraging data.** The center-of-mass coordinates, as well as intensity-related measures such as mean intensity, maximum intensity, and centerline of the *C. elegans* pharynx, were extracted using PharaGlow[24]. Using a peak-detection algorithm, intensity peaks of the GCaMP8f indicator were annotated and used to calculate a pumping rate, similar to the process described in refs. 19,24.

**Drosophila maintenance and strains.** All fly stocks were maintained at 25 °C. Flies expressing MEF2-Gal4 (Expresses GAL4 in muscle cells, likely BDSC_27390), UAS-mGerry (GCaMP6m fused to mCherry under the control of UAS, BDSC_80141) were provided by E. Heckscher. The genotype of the flies in the experiments was w*, MEF2-Gal4, UAS-mGerry (see Table 1).

**Single *Drosophila* larvae chemotaxis assay.** All chemotaxis assays involved early and late second-instar larvae tested during the day. The room temperature was maintained at around 20 °C, and the relative humidity of 30 ± 6%. Chemotaxis assays were performed on 15 cm unseeded imaging NGM plates. At one end of the plate, 20 μL of undiluted commercial apple vinegar (REWE, Germany) was deposited. A single larva was positioned within a droplet of distilled water approximately 3 cm from the vinegar drop. Their locomotor movement was videotaped for 5 min at 1× magnification with 20 ms exposure using the dual-color configuration and a sampling rate of 19.6 frames per second for 5 min.

### *Drosophila* crawling analysis

Dual-color images were analyzed with PharaGlow[24]. The segmentation was adapted to reliably detect the larvae (Supplementary Fig. 3), virtually straighten the animal, and extract kymographs of intensity along the anterior-posterior axis. The red channel (R(t)) was used to correct motion artifacts of the calcium signal (G(t)) using the same approach as described in ref. 14. In brief, for each location on the anterior-posterior axis (L), we estimate α(L), such that the corrected signal is $F_{corr} = (G - \alpha R) - \langle G - \alpha R \rangle$, with α(L) minimizing $\sum (G(t) - \alpha R(t))^2$.

### Calculating the resulting magnification

For fixed focal distance objectives (e.g., 50 mm Yongnuo, 12 mm EO), the magnification is given by $M = f_{TL}/f_{OBJ}$. As the microscope uses a non-standard tube lens ($f_{TL, GlowTracker} = 50$ mm), the magnification of the attached objectives changes from the nominally stated magnification. For commercial objectives such as the Olympus 10× and Olympus 20×, the magnification can be calculated as $M_{effective} = \frac{f_{TL}}{fOlympus_{TL}} M_{nominal}$. For an Olympus 10× objective with $fOlympus_{TL} = 180mm$, $f_{TL} = 50mm$, and $M_{nominal} = 10x$, this results in $M_{effective} = 50/180 * 10 \simeq 2.78$.

*Predation Assay* Young *C. elegans* larvae (SJ4103) were washed with M9, passed through a 20 μm filter twice, centrifuged, and deposited on an unseeded 10 cm NGM imaging plate by pipetting 4 μL of the worm pellet. The plate was left undisturbed for at least 1 h to allow the larvae to evenly distribute on the plate. Following a 2-h starvation period, six washed young adult *Pristionchus pacificus* (JWL27) predatory hermaphrodites were introduced to the plate. Animals were imaged with a 16 mm objective at 30 fps using the dual-color microscope. Two neutral density filters with an optical density (OD) of 0.6 and 1.0 were added to reduce the intensity of the red imaging channel, as the fluorescence of *Ppa-myo-2p::tagRFP* of the adult predator was stronger than the label of the prey, making the adjustment of the dynamic range of the camera difficult for both colors simultaneously.

### Statistical analysis of prey signal

The recordings were analyzed with a custom Python script, incorporating the imaging analysis tool PharaGlow[24], to extract the pharynx centerline, center of mass, and additional metrics of the predator (*P.*

*pacificus*). The coordinates of the recordings were transformed into the stage space and used for velocity calculation. Following this, the behavior of the predators was analyzed with a machine-learning-based model published to detect behavioral states[49]. The model uses features that contain information about the pharyngeal contractions, velocity, pharyngeal curvature, as well as frequency and positive and negative lags of the features. Further, the extracted centerline was used to measure the prey signal in the green channel. For this, a 34 μm wide circular mask, centered around the anterior end of the predator's centerline, was used. The prey signal was calculated from the raw GFP signal as the ratio: $preysignal = \left( GFP_{95percentile} - GFP_{5percentile} \right)/GFP_{5percentile}$. To examine behavior and prey signal around the onset of predatory biting, the tracks were aligned at the onset of predatory biting events. To analyze the rise of prey signal at biting onset, the prey signal of each track was normalized to a baseline defined as the mean of the prey signal between 15 and 5 s before bite onset, as $preysignal(\%) = (preysignal - baseline)/baseline$. Statistics were performed using a paired t-test, comparing the time ranges of 15 to 5 s before bite onset and 0 to 15 s.

### Calcium imaging of PLM

*C. elegans* young adults expressing *mec-17p::GCaMP8f::mCherry2* were transferred 15 min before recording to an NGM imaging plate, and shortly before the recording, the agar was cut and transferred to the recording setup. The NGM plates were seeded with 50 μL of OP50 the evening before the experiment and allowed to grow overnight. Animals were recorded in a dual-color microscope, as described in Supplementary Data 1. A tube lens of type Yongnuo 50 mm and an objective of type EO 12 mm were used, resulting in an effective magnification of 4.1× (Supplementary Table 2). The red channel was used for tracking the animal, and the intensity of the red channel was attenuated using an OD = 1 neutral density filter to account for the brightness difference between the two channels.

### Touch stimulation during calcium imaging

Animals were stimulated with a sinusoidal wave at 630 Hz and 20 Vpp using a piezo buzzer as described in ref. 19. The stimulation protocol had five repeats of 1 s with an interstimulus interval of 30 s and a pre-stimulation period of 30 s. The stimulation was controlled via a custom MATLAB script and an NI DAQ board (NI, TX, USA).

Extraction of the calcium signals was performed with custom Python scripts. In brief, after cropping the images to a 50×50 px region of interest, the background (defined as the 50th percentile of pixel intensity in every 100th frame) was subtracted from the red channel. Neurons were detected by blurring with a Gaussian filter (standard deviation = 0.3), and the neuron mask was found using thresholding. The mask was then used to extract the raw signal from the reference and signal images. For ratiometric analysis, ΔR/R0 was calculated as

$$R = \frac{\frac{F_{sig}}{F_{sig} + 1}}{\frac{F_{ref}}{F_{ref} + 1}} \quad (1)$$

$$\frac{\Delta R}{R0} = \frac{R - \bar{R}}{\bar{R}} \quad (2)$$

Velocity was calculated directly from stage coordinates, which were recorded at 30 fps.

### Statistical analysis of calcium imaging

For plotting and statistical analysis, the traces were aligned at the stimulus onset. For the statistical analysis, we used a dependent t-test for paired samples to compare the mean value of the prestimulus interval (from −10 s to 0 s aligned to the stimulus) and the stimulus interval (0 s to 1 s) within one stimulus repeat. For Fig. 3E, F, the repeats were

filtered according to reversal events occurring within 2 s of stimulus onset. Reversals were defined based on the method described in ref. 72 by detecting sharp angular changes in the trajectory.

The effect of reversals on the mean post-stimulus calcium signal was tested using a mixed linear model with PLM activity as the dependent variable and reversals as the independent variable, categorical, while accounting for random effects between worms. The analysis was performed with the 'statsmodels' package in Python.

## Tardigrade maintenance

Tardigrades were kept at room temperature (20 °C) in mineral water (Volvic) with a suspension of *Chlorella vulgaris* grown in BG11 medium added regularly as a food source. Animals of approximately the same size were selected from a culture plate for tracking (see Table 1).

## Brightfield tracking of tardigrades

Tardigrades were observed on 6 cm unseeded 1.5% agarose plates containing Volvic water. Locomotion was recorded using a 12 mm EO objective at 4.1× magnification in brightfield mode, employing a single-color configuration with red LED illumination (650 nm) and a 50/50 image splitter instead of a dichroic mirror. Images were captured at 15 frames per second with an exposure of 300 μs. The tardigrades' center of mass was tracked in the X−Y plane for 10 min using the GlowTracker software.

## Pose estimation of tardigrades

The pose estimation was done with DeepLabCut (3.0.0rc4). Training was done sequentially, with more frames extracted in later training iterations to improve pose estimation. We used 12 recordings not included in the latter analysis for training a new DeepLabCut (DLC) model. In the last iteration, we extracted 5 more frames from all 10 recordings used during analysis to further improve pose estimation for those videos. Because the recorded.tiff files cannot be directly imported into DLC. We saved the recordings as.avi files without compression. From those, we extracted and labeled a total of 354 frames. Labeling was done as shown in Fig. 6B: we labeled the eyes (left and right), all four leg pairs (each left and right), and two additional points at the anterior and posterior ends of the body.

For training, we used 95% of the labeled data. We trained a ResNet-50-based neural network with default parameters for 200 training epochs. This was sufficient to achieve a root mean squared error on the test set of 3.12 μm (5.29 pixels), while the training error was 1 pixel.

For analysis, video recordings and pose estimations were aligned with the stage coordinates. When the likelihood of the pose estimation of any body part fell below 70%, the position of the given body part was linearly interpolated. The tardigrades' center of mass was calculated as the mean of all tracked body parts. To extract the position on the anterior-posterior axis of all body parts, the positions of the body parts were transferred into a straightened coordinate system. Here, the front, rear, and the mean between all leg pairs were used to calculate centerline points that were equally distributed in space. For each body part, the closest point on the centerline was determined and used as the anterior-posterior position of the given body part. Moreover, the distance of the body part to the centerline was used as the lateral distance. Swings were extracted automatically using the anterior-posterior position over time. The onset of swings was defined as the peak of the first derivative (speed) and the offset as the trough of the second derivative (maximum deceleration).

To estimate the correctness of the automatic swing detection, swings were manually labeled in a small fraction of the recordings. Specifically, 10 s of 5 recordings were randomly extracted, manually annotated, and compared to the automatic detection. To further compare the swing detection with previously published experiments, the duty factor was calculated as the proportion of stance duration to stride duration. For each animal, Spearman correlation coefficients were estimated individually. The Spearman coefficients r were then averaged using the Fisher-z transform, and the reported *p*-values were combined using the Fisher method.

## Gait identification of tardigrades

To identify gait types in the videos, we deployed a machine learning approach. For this, we first partitioned the time series of the anterior-posterior position of all body parts into small snippets, representing approximately one gait cycle. We determined the length of a gait cycle by the appearance of two swings in the same body part, with a cycle ending before the second swing started.

All snippets were interpolated to contain 15 sampled timepoints. We then calculated the distance between all snippets using dynamic time warping, considering all four leg pairs (or all three anterior leg pairs in a supplementary analysis (Supplementary Fig. 5). The pairwise distance could then be used to embed the individual gait cycles into a two-dimensional embedding using UMAP, similarly to refs. 49,73. Next, we performed clustering on this embedding using Agglomerative Clustering and calculated the barycenters of the clusters, showing the representative sequence of each cluster, as well as the mean. The clusters were visually analyzed and compared to previously described gaits. Lastly, the distribution of cms velocity and track angle of each cluster was calculated.

## 3D model of the microscope

The 3D model of the single and dual-color microscope was created using the Fusion 360 software under a license provided by the Max Planck Digital Library.

## Reporting summary

Further information on research design is available in the Nature Portfolio Reporting Summary linked to this article.

# Data availability

The data underlying this manuscript are shown in the figures and supplementary tables. Extended behavioral and imaging data are available at https://osf.io/rpfny/. Code for generating the figures is deposited under https://github.com/scholz-lab/Macroscope-Paper.

# Code availability

The Glowtracker software is available on GitHub https://github.com/scholz-lab/GlowTracker and installable via the Python package manager pip as "glowtracker". Further documentation and instructions are available at https://scholz-lab.github.io/GlowTracker. At the time of publication, the version of GlowTracker was glowtracker 0.11.0.

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

## Acknowledgements

We thank Ellie Heckscher and Matthieu Louis for the reagents. We thank Louis Wolinski for assistance with the 3D models. We thank members of the laboratory of James Lightfoot (Fumie Hiramatsu and Desiree Goetting) for testing our documentation by building a microscope and providing valuable feedback. This research was supported in part by the National Science Foundation under Grant No. NSF PHY-1748958 and the Gordon and Betty Moore Foundation Grant No. 2919.02 (MS). The project iBEHAVE (M.S. and L.B.) has received funding from the program "Netzwerke 2021", an initiative of the Ministry of Culture and Science of the State of North Rhine-Westphalia. The sole responsibility for the content of this publication lies with the authors. Part of this work was funded through the BABots project. The BABots project has received funding from the Horizon. Europe, Pathfinder European Innovation Council Work Program under grant agreement No 101098722 (MS). Views and opinions expressed are, however, those of the authors only and do not necessarily reflect those of the European Union or European Innovation Council and SMEs Executive Agency (EISMEA). Neither the European Union nor the granting authority can be held responsible for them.

## Author contributions

Conceptualization: L.A., M.S.; Data curation: E.R., L.B.; Funding acquisition: M.S.; Investigation: E.R., L.B., T.S., B.S., L.A., M.S.; Methodology: E.R., T.S., L.A.; Project administration: M.S.; Resources: L.A.; Software: T.S., B.S., M.S.; Supervision: M.S.; Validation: L.A.; Visualization: E.R., L.B., T.S., M.S.; Writing—original draft: E.R., M.S.; Writing—review & editing: E.R., L.B., L.A., M.S.

## Funding

## Competing interests

The authors declare no competing interests.
