## [Transparent Peer Review file · Nature Communications]

A modular multi-color fluorescence microscope for simultaneous tracking of cellular activity and behavior

Corresponding Author: Dr Monika Scholz

Version 0:

Reviewer comments:

Reviewer #1

(Remarks to the Author)

The authors report the development of a low-cost, accessible, customizable microscope and its application in the analysis of an impressive array of different animals and experimental conditions. Their tracking microscopy enables quantification of behavior in freely moving, unconstrained animals, and dual-color fluorescence microscopy enables real-time recording of neuronal activity, and multiple types of targets (e.g. in differently labeled tissues or differently labeled individuals). These results will be valuable to (1) researchers who are interested in prototyping new analysis paradigms and collecting preliminary data without significant financial investment, (2) researchers interested in quantifying behavior in freely-moving, unconstrained animals, and potentially collecting simultaneous cellular-level activity data, and (3) educators or researchers at education-focused institutions who desire relatively sophisticated analyses at an affordable price, enabling accessibility for a large population of students.

The authors stress the accessibility and affordability of their microscope hardware design. This accessibility is enhanced by a detailed set of assembly instructions and parts at (<https://scholz-lab.github.io/GlowTracker/software/software.html>). The affordability is documented on that site as well as in Table S2 of the manuscript.

To enable effective use of this accessible hardware, the authors have developed software that is operable on all major personal computer platforms. Ease of use is emphasized by the development of a Graphical User Interface (GUI), by extensive instructions and documentation on the accompanying website, and by thoughtful characteristics that facilitate downstream analyses (e.g. saving dual-color videos as image stacks wherein color balance is already set, thereby minimizing post-processing steps.)

The authors then provide several examples showcasing different ways their hardware and software can be used. Specifically, they measure:

1. Peristaltic muscle contractions in *Drosophila* larvae, using real-time individual tracking along with GCaMP6 signal quantification
2. Activity of touch responsive neurons in forward and reverse escape behaviors of *C. elegans*, while showing that posterior touch sensing neurons are more active during forward escapes
3. Interactions between predatory nematodes and their (nematode) prey
4. Stability of their system during long term (1h) foraging behaviors in freely-moving *C. elegans* adults
5. Whole animal movement and characteristics of individual leg motions in the tardigrade *Hypsibius exemplaris* moving freely over large distances and long times.

While none of these test cases generated particularly new or exciting data, they nicely show the wide range of locomotor behaviors and characteristics of neuronal or muscular activities that are quantifiable using their setup.

Specific comments on TEXT:

Line 18: Consider rewording "...simultaneous calcium imaging of neurons and behavior...". As written, it sounds like behavior is assayed via calcium imaging.

Lines 209-230 '... and the environment' is too general and does not really capture what the authors are measuring here. The authors are measuring predator-prey interactions. The more impressive points are that they are tracking two different kinds of animals simultaneously, that these animals have different priorities and behaviors, and that they are able to predict different behaviors of the whole animal based on the structure of a single organ.

Lines 215-220: The 'previously-developed ... machine-learning model to identify behavioral states of the predator' should be cited if published, or it should be clarified that this paper is the first presentation of the model. Lines 221-230: more context needs to be provided to clarify what is meant by 'pumping rates' and 'prey signal', so that the non-specialist reader can have a clearer picture of the assay. In addition, clarify that 'velocity' refers (presumably) to the speed of the predator species.

Line 259 - consider changing 'non-model' to 'tardigrades' here and elsewhere in the manuscript. There is a growing community of researchers who are striving to use tardigrades to model many aspects of biology. In addition, 'Non-model' is becoming a less-useful descriptor in an age of high-throughput, inexpensive genome sequencing.

Lines 260-285 Tardigrade tracking. One key advance in this study is the use of automated pose assessment with DeepLabCut. It would be useful to more fully explore the accuracy of this approach, especially in 'occluded or out-of-focus' areas. In addition, it would be helpful to know what fraction of each particular video contained usable data. These limitations would benefit from a deeper, more thorough analysis, as they can potentially be impacted by differences in image acquisition, and may lead to problematic datasets.

For example, in 6F&G, the difference in video image characteristics are quite striking between these two tardigrades. For example, the individual claws of the tardigrade in (F) are much more readily visible than those in (G). It would be useful to include a discussion of potential differences in automated stride tracking depending on differences in contrast, z-level focus, or other differences in image acquisition.

Of particular note: the data shown in G look odd - there appear to be periods of time (e.g. ~226.5, ~227, ~227.5, 230 seconds) where *all* legs appear to be in swing phase, which is probably not possible in a walking tardigrade, and/or may indicate the the tardigrade is not in fact walking but has lost contact with the substrate (in which case a 'gait' is not really the appropriate term or concept, and thus a 'galloping gait' would not be relevant).

Alternatively, the problematic swing timing in G could have been produced by the automated peak detection using `scipy.find_peaks`. It would be useful to compare manually-called swings and stances with those produced by the automated peak detection, certainly in the two examples shown in the figure and probably for additional examples as well, to get a more complete measure of the accuracy of automated stride calling.

In all, these potential mis-calls in swing timing are a significant issue and should be addressed before publication.

The data in F is also odd (to a much lesser degree), as in a canonical tetrapod gait, an L1 swing would be expected at about 187-188 seconds. Was there a real swing that was missed by the automated pose detection or by the automated peak calling? Occasionally a tardigrade will in fact 'miss' an expected swing, but the doubts raised by potential problems in the automated swing calling suggest that the 'absent' swing may stem from a problem with sensitivity or accuracy of stride detection.

Lines 274-277. Related to the above: the authors should clarify their gait terminology. For example, a 'tetrapod' gait in tardigrades generally describes a pattern in which four of the anterior six legs are in stance phase, while two are swinging. A 'canonical' tetrapod gait occurs when the swinging legs are in adjacent segments but on opposite sides of the body, while a galloping gait occurs when swinging legs are in the same segment. Any gait with more than two legs swinging (within the anterior three pairs) would not be a tetrapod gait.

In line 281, the authors state that they 'capture more strides per animal than previous studies' - this statement needs justification and comparison of the real numbers reported here and in the previous studies. Importantly, these comparisons should only use the high quality portions of video that were actually used in the automated pose estimation, excluding the "... large fraction of frames where the tardigrades were either not in focus, in an undesired position, or obscured by algae." (lines 466-467).

The authors further state that their extended data collection may enable 'the observation of gait transitions' (line 282). Such transitions have already been observed (in refs 51 and 52), and would not be a new avenue of analysis opened by this particular study. Discrete gait transitions are not readily observed in tardigrades; rather, they smoothly transition between coordination patterns (see ref 51).

Line 449 - change to "Brightfield tracking of tardigrades"

Comments on FIGURES and FIGURE LEGENDS and TABLES

In the 1D legend, the top-to-bottom order of the magnification values in the images do not appear to correspond to the descriptions in the legend. Also, the 'magnification' values are hard to square with the scale bar lengths shown in the images. Consider just noting, for example, 3.125X rather than 3.125x magnification. Or (perhaps better) let the scale bars speak for themselves in these and in other images in the manuscript.

Figure 2 title: add the species and italicize that (i.e. rather than 'Larvae').

Figure 2F the inset is a bit confusing because the lower track that veers off to the right appears to be from a later(?) time period than the rest of the inset. Consider omitting this lower track from the inset.

Figure 2F legend: consider 'example' rather than 'exemplary'.

Figure 3C, D legend could be organized more logically (e.g. put the **** $p < 0.0001$ with D). It would be useful to add information in the legend (also for 3E) about what specifically is being compared in the statistical tests.

Figure 3F legend: include more detail about the statistical test used.

Figure 4 legend would benefit from more stand-alone context and explanation regarding 'pumps' and (especially) 'prey signal'. Figure 4E states that 'prey signal' was detected as in C, but no information is given in the legend about how prey signal is detected in C, or what the term 'prey signal' means.

Figure 4B legend is incomplete ("...predicted behavioral states using the model published in .")

Figure 4D it may be useful to partition these 43 tracks according to the 8 predators, in order to compare variation between the predators.

Figure 5A is missing the 'A'.

Figure 5C it would be useful to include an outline (dotted white line?) around the entire *C. elegans* individual, for context. As before, the magnification value in 5C is a bit confusing in combination with the scale bar on the image (the image as shown appears to be magnified more than 3.125x). Consider just including the scale bar, or the specific objective used (EO 16 mm?).

Figure 5E is missing the 'E'. It would be useful to have Y-axis values shown here (e.g. animal 1, animal 2, etc.)

Figure 5G - would be useful to have animal labels here too (as in 5C).

Figure S1B - What is the significance (if any) of the outlier point at ~33Hz image acquisition?

Table S2: To put 'affordability' in real world context, it would be useful to add the total costs in Table S2, and report these values in the main text, and compare different configurations (brightfield, single color fluorescence, dual color fluorescence).

(Remarks on code availability)

The Python code is well-commented, neat, and expertly constructed. A 'readme' file is available with installation and usage instructions. I did not download (or run) the code, as it primarily controls hardware I do not possess, and analysis of videos acquired from this hardware.

Data is reported to be at osf.io/rpfny ... when I visited this site on 3 June 2025, it was not accessible - it said "You Need Permission".

Reviewer #2

(Remarks to the Author)

Synopsis:

A modular multi-color fluorescence microscope for simultaneous tracking of cellular activity and behavior reports the development of a modular tracking microscope (or macroscope) that can incorporate neurobiological data using real-time recording of fluorescent indicators. This work fits an ongoing pursuit by a number of labs and follows publications such as Arous 2010 (10.1016/j.jneumeth.2010.01.011), Venkatachalam 2016 (10.1073/pnas.1507109113), Nguyen 2016 (10.1073/pnas.1507110112), Voleti 2019 (10.1038/s41592-019-0579-4), among others that the authors cite. The microscope reported here is cheaper than many of the others, and the authors have shown it is capable of tracking other small invertebrates like tardigrades and fly larvae (though this has also been reported before by Huang 2024 (10.1016/j.bbrc.2024.150290)). Though no novel capabilities are reported, the modularity, affordability, and open-source nature of the microscope and software make it a welcome addition to the field.

Major manuscript comments:

Since a primary use-case is behavioral tracking, I think more could be included in the text about caveats associated with environmental control, lack thereof, or the potential for adding it, and how the imaging environment could impact behavior. For example, it isn't entirely obvious that the light source is filtered through a slit (only mentioned in figure captions), which I assume means that the rest of the arena is in the dark (the Getting Started page of the documentation suggests this). The effects of a light gradient in the behavioral arena are difficult to predict, even if the tracked organism itself is always illuminated.

Can other types of environment control be added?

Lines 465-470 could use clarification. A subset of videos was used for training, but it's unclear what subset. The next sentence reads as if the training set included best-case and edge-case (poor) frames, but then the next sentence seems to walk that back by suggesting only best-case frames were used for analysis. If edge-case frames were used for training, why

were they ignored during prediction? What proportion of frames included a predicted model that surpassed a confidence threshold? Since a feature of the system is its potential for use with non-model organisms, it's important to understand just how robust the imaging and tracking is in cases where perfection is not possible.

What is the largest size of arena that could be potentially be used? Line 112 states that the motorized stage can move 150 mm – is that in both X and Y? Have the authors used an arena this large?

Minor manuscript comments:

The authors need to decide if it's a microscope (manuscript) or a macroscope (documentation).

Throughout it seems that "stage" is used to mean "camera," since it's the camera that is moving, not the stage that the plate is set upon. This needs fixing or clarifying.

(Remarks on code availability)

I reviewed the GitHub repo and documentation website but did not closely examine the code for tracking implementation.

More should be said about the data that comes out of GlowTracker, including file formats, compression, typical file sizes, etc. The tardigrade videos were seemingly imported to DeepLabCut – is cross-platform integration of the output files straightforward? Do users get access to both videos (i.e., MP4 or AVI) and coordinate data (CSV, HDF5, other)?

I cannot find documentation on initiating tracking of a specific animal. The tracking_explanation page includes a lot of technical information that is interesting but superfluous for most users. In a multi-animal experiment, such as the examples on that page or the predator vs. prey example in the manuscript, how does the user select the animal to track? Is tracking fidelity maintained upon animal collisions/overlap? This is an important question to answer about the experiment highlighted in Figure 4.

I was able to install and start the GUI on a Mac Studio (Apple Silicon), though unable to test the functionality without the camera. I will note that mixing conda environments and pip installs is not best-practice. I highly recommend uv for package management, or at least synchronicity with pip and venv or virtualenv. For potentially broader adoption by non-technical users, the software could be containerized for launch with Docker Desktop bundled with a launcher like PyInstaller. Linux and Windows are mentioned in the documentation, but I can't find anywhere that lists OS's that have been confirmed to work.

Reviewer #3

(Remarks to the Author)

The objective of the paper was to build a simple microscopy tool consisting features like live tracking of small model organisms (C elegans, fly larva etc etc) with fluorescence set up (single and dual colour) as well as bright field microscopy (tardigrade). They also combined some quantitative measurement of neural activity with genetically encoded calcium sensor (GCaMP) within the neuron of transparent models. They also suggest that the system is easy to assemble by non-experts.

They have given some experimental examples in the nematode C. elegans, Drosophila larvae and tardigrade.

According to me it is an interesting paper discussing a microscopy tool development, which is helpful for researcher in general. However, I would like to point out that there are a number of papers which deals with similar methodology including few in worm. Every microscopy techniques and assays they have used are already established before and they have provided reference too. I am not sure whether this is appropriate for Nature Communication as I don't see any new findings as well as these tools are already used by researchers in various models. I agree that these methodology are useful, but the authors at least could not convince how these microscopy-tools would solve any question that is unanswered as of now.

I am also listing few specific questions below in regard to the presented data in this manuscript.

1) This paper majorly focuses on cost-effective and vibration free tracking system for Sample with Lightsource moving while tracking. Their epifluorescence system cost around ~14000 Euros where else similar dual epifluorescence vibration free system WormsPY (2024) cost around ~12300 Euros with easily available parts. See the reference # 53.

2) Figure-1: You have shown some brightfield images of tardigrade and larvae in upcoming results in the same microscope. Can you put light path of brightfield also in Figure 1 ? like you have done red and green.

Suggestion: In figure 1D Can you show GCaMP and mCherry in separate channel like you did in Fig 3B? It is easier to see

3) Figure 2: In this result section authors describe, how fly larvae crawls towards an attractive odours. They tracked the larvae with fluorescently labelled neuron. In figure-D-E: They show how the Odors experienced by the successful (blue) and unsuccessful (red) larvae over time. Not sure why one has to use fluorescent larvae for this as simple brightfield is sufficient to capture and track the movements.

Also, it's not clear from this the relationship between neural activity and successful vs unsuccessful trial. Since GCaMP sensor is present in the sensory neuron, one expects some neural activity could be monitored in this case.

1F-G: Does there is GCaMP6 activity differ in muscle of motile larvae vs non-motile larvae? That comparison kymographs will be interesting to see.

4) Figure #3: It's not clear where the piezo is kept on NGM plate. Please have a clear illustration.

I would also like to see the GCaMP recording from ALM neuron as well, which might be responsible for the reversal behaviour in response to mechanical cue.

I would also like to see more details of the behavioural quantification in response to mechanical stimuli. What percentage of worms show reversal vs forward movement?

How can one rule out the effect of other mechanosensory neurons such as PVD in this experiment?

The authors should test the behavioural response in both *mec-3* (null for large force) and *mec-4* (null for gentle force) mutants.

5) Figure 4: Visualising animal and environment interaction

Query: Can you show the brightfield video frames of prey-predator interaction. With just fluorescence imaging, it is not clear

6) Figure 5: Modular single-color design enables long-term tracking.

Query: Is there any condition where pharyngeal pumping significantly changes? It can validate the tool further for long-term assay

For example *itr-1* mutant (lacking IP3 signalling).

7) Figure 6: Bright-field tracking for non-model species

Query: Only 9 videos were used to train the model, How many tardigrades were there in each video?

8) Also, it would be nice if they can comment and give details whether their system would allow optogenetic manipulation (Channelrhodopsins) of neurons and simultaneous tracking of behaviour or imaging of GCaMP response in neurons. This is another important component needs to be clear.

(Remarks on code availability)

The authors did a good job in detailing the methods and codes. The code provide the README file and instructions are there for installation.

I was able to install and run the code.

Version 1:

Reviewer comments:

Reviewer #1

(Remarks to the Author)

The authors develop a low-cost accessible system, including both hardware and software, for tracking and quantification movement, at impressive scales in both space and time. They demonstrate the performance of their hardware and software by tracking a variety of species, including whole-animal tracking via fluorescent markers, real-time calcium imaging, interactions between species, and extendability to multiple kinds of moving animals. The inclusion of a detailed parts list with prices is especially invaluable, as it will allow readers to compare with alternative systems and to modify the parts to suit their needs. In all, the authors describe a set of tools that will be of interest to researchers looking to quantify locomotor behaviors in real time. I appreciate the honest and thorough efforts the authors have made to address reviewer concerns, and I especially appreciate the completely new analyses of tardigrade stepping patterns. I still have a few misgivings about the accuracy of the automated step tracking, and the determinations of stances vs. swing phases, but the inclusion of the anterior-posterior positioning of each leg allows the reader to evaluate the accuracy of the phase determination on their own. The authors also compare the accuracy of automated tracking with manual tracking. These are minor points, however, as the paper is more about hardware and software development than it is about the intricacies of tracking leg motion in a particular species.

Some suggestions that would further strengthen this section of the paper:

1. As an additional measure of accuracy of automated stance-swing annotations, I suggest that the authors compare the duty factors obtained in their dataset (ideally in a plot vs. speed) with those of previous studies on tardigrades (refs 53 and 54). This comparison would not require additional experiments - just extraction of additional parameters from their existing data. This analysis would help determine if there are systematic errors in their phase-calling (e.g. if swing phases are too long).
2. Also, I would suggest reversing the colors so that swing phases are shown in black, which seems to be the convention (see for example Mendes et al. 2013, DOI: 10.7554/eLife.00231, Wosnitza et al. 2013, doi:10.1242/jeb.078139, and others).
3. Include a legend with the colors for swings and stances for Figure 6F and 6G.
4. For Figure 6F, I would also recommend reorganizing the legs according to their side (first) and then segment, as done in the aforementioned studies in *Drosophila*, and the tardigrade walking studies cited (refs 53 and 54).

Finally, double check citations throughout the paper. For example line 334: "The normalized mean anterior-posterior position of the different cycles show similarities with gaits previously described in tardigrades (Fig. 6G) (56, 57)" ... but 56 and 57 are studies in *Drosophila*.

(Remarks on code availability)

Code is well-documented. The README file provides easily-understood intructions for installation and use.

Reviewer #2

(Remarks to the Author)

I thank the authors for their thoughtful responses to my comments.

(Remarks on code availability)

The README and documentation are sufficient. I was able to install and run the code.

Reviewer #3

(Remarks to the Author)

The authors have addressed all the questions and the revised manuscript is suitable for publication.

(Remarks on code availability)

The authors did a good job in detailing the methods and codes. The code provide the README file and instructions are there for installation.

I was able to install and run the code.

REVIEWER COMMENTS

Reviewer #1 (Remarks to the Author):

The authors report the development of a low-cost, accessible, customizable microscope and its application in the analysis of an impressive array of different animals and experimental conditions. Their tracking microscopy enables quantification of behavior in freely moving, unconstrained animals, and dual-color fluorescence microscopy enables real-time recording of neuronal activity, and multiple types of targets (e.g. in differently labeled tissues or differently labeled individuals). These results will be valuable to (1) researchers who are interested in prototyping new analysis paradigms and collecting preliminary data without significant financial investment, (2) researchers interested in quantifying behavior in freely-moving, unconstrained animals, and potentially collecting simultaneous cellular-level activity data, and (3) educators or researchers at education-focused institutions who desire relatively sophisticated analyses at an affordable price, enabling accessibility for a large population of students.

The authors stress the accessibility and affordability of their microscope hardware design. This accessibility is enhanced by a detailed set of assembly instructions and parts at (<https://scholz-lab.github.io/GlowTracker/software/software.html>). The affordability is documented on that site as well as in Table S2 of the manuscript.

To enable effective use of this accessible hardware, the authors have developed software that is operable on all major personal computer platforms. Ease of use is emphasized by the development of a Graphical User Interface (GUI), by extensive instructions and documentation on the accompanying website, and by thoughtful characteristics that facilitate downstream analyses (e.g. saving dual-color videos as image stacks wherein color balance is already set, thereby minimizing post-processing steps.)

The authors then provide several examples showcasing different ways their hardware and software can be used. Specifically, they measure:

1. Peristaltic muscle contractions in *Drosophila* larvae, using real-time individual tracking along with GCamp6 signal quantification
2. Activity of touch responsive neurons in forward and reverse escape behaviors of *C. elegans*, while showing that posterior touch sensing neurons are more active during forward escapes
3. Interactions between predatory nematodes and their (nematode) prey
4. Stability of their system during long term (1h) foraging behaviors in freely-moving *C. elegans* adults
5. Whole animal movement and characteristics of individual leg motions in the tardigrade *Hypsibius exemplaris* moving freely over large distances and long times.

While none of these test cases generated particularly new or exciting data, they nicely show the wide range of locomotor behaviors and characteristics of neuronal or muscular activities that are quantifiable using their setup.

Specific comments on TEXT:

Line 18: Consider rewording "...simultaneous calcium imaging of neurons and behavior...". As written, it sounds like behavior is assayed via calcium imaging.

We have rephrased it as "...calcium imaging of neurons in behaving animals..."

Lines 209-230 '... and the environment' is too general and does not really capture what the authors are measuring here. The authors are measuring predator-prey interactions. The more impressive points are that they are tracking two different kinds of animals simultaneously, that these animals have different priorities and behaviors, and that they are able to predict different behaviors of the whole animal based on the structure of a single organ.

We appreciate the positive comments and have expanded our discussion of the importance and significance of these data. We thank the reviewer for suggesting more impactful language.

Lines 215-220: The 'previously-developed ... machine-learning model to identify behavioral states of the predator' should be cited if published, or it should be clarified that this paper is the first presentation of the model.

We appreciate the reviewer's comments and cite the corresponding recently accepted publication.

Lines 221-230: more context needs to be provided to clarify what is meant by 'pumping rates' and 'prey signal', so that the non-specialist reader can have a clearer picture of the assay. In addition, clarify that 'velocity' refers (presumably) to the speed of the predator species.

We now edited the figure and the corresponding text to make both more accessible to non-specialist readers and replaced 'velocity'.

Line 259 - consider changing 'non-model' to 'tardigrades' here and elsewhere in the manuscript. There is a growing community of researchers who are striving to use tardigrades to model many aspects of biology. In addition, 'Non-model' is becoming a less-useful descriptor in an age of high-throughput, inexpensive genome sequencing.

We appreciate and agree with the reviewer's comment and now cite a paper of the tardigrade community highlighting its role as a valuable emerging model organism.

Lines 260-285 Tardigrade tracking. One key advance in this study is the use of automated pose assessment with DeepLabCut. It would be useful to more fully explore the accuracy of this approach, especially in 'occluded or out-of-focus' areas. In addition, it would be helpful to know what fraction of each particular video contained usable data. These limitations would benefit from a deeper, more thorough analysis, as they can potentially be impacted by differences in image acquisition, and may lead to problematic datasets.

We appreciate the reviewers interest in our automatic pose estimation and swing detection. We agree that the previous dataset had limitations. We recorded a larger new dataset, and compared automatic swing detection with manual annotation. We hope this alleviates all previous concerns.

For example, in 6F&G, the difference in video image characteristics are quite striking between these two tardigrades. For example, the individual claws of the tardigrade in (F) are much more readily visible than those in (G). It would be useful to include a discussion of potential differences in automated stride tracking depending on differences in contrast, z-level focus, or other differences in image acquisition.

Of particular note: the data shown in G look odd - there appear to be periods of time (e.g. ~226.5, ~227, ~227.5, 230 seconds) where *all* legs appear to be in swing phase, which is probably not possible in a walking tardigrade, and/or may indicate the the tardigrade is not in fact walking but has lost contact with the substrate (in which case a 'gait' is not really the appropriate term or concept, and thus a 'galloping gait' would not be relevant).

Alternatively, the problematic swing timing in G could have been produced by the automated peak detection using `scipy.find_peaks`. It would be useful to compare manually-called swings and stances with those produced by the automated peak detection, certainly in the two examples shown in the figure and probably for additional examples as well, to get a more complete measure of the accuracy of automated stride calling.

In all, these potential mis-calls in swing timing are a significant issue and should be addressed before publication.

We appreciate the reviewer's detailed comments regarding our gait analysis. We have completely redone the experiments and analyses, and have also established manual swing annotation. We compare the automatic and manual annotation and report performance metrics. We appreciate the reviewers' patience and hope our new figure resolves all the open questions.

The data in F is also odd (to a much lesser degree), as in a canonical tetrapod gait, an L1 swing would be expected at about 187-188 seconds. Was there a real swing that was missed by the automated pose detection or by the automated peak calling? Occasionally a tardigrade will in fact 'miss' an expected swing, but the doubts raised by potential problems in the automated swing calling suggest that the 'absent' swing may stem from a problem with sensitivity or accuracy of stride detection.

We appreciate the reviewer's comments regarding the two examples. We completely redid the experiments and analyses. Our new analysis includes a data-driven exploration of gait types, which relies not on strides, but directly on the pose estimation, the root mean squared error of which we report.

Lines 274-277. Related to the above: the authors should clarify their gait terminology. For example, a 'tetrapod' gait in tardigrades generally describes a pattern in which four of the anterior six legs are in stance phase, while two are swinging. A 'canonical' tetrapod gait occurs when the swinging legs are in adjacent segments but on opposite sides of the body, while a galloping gait occurs when swinging legs are in the same segment. Any gait with more than two legs swinging (within the anterior three pairs) would not be a tetrapod gait.

We apologize for our incorrect use of the gait terminology and have now adapted the proper terms.

In line 281, the authors state that they 'capture more strides per animal than previous studies' - this statement needs justification and comparison of the real numbers reported here and in the previous studies. Importantly, these comparisons should only use the high quality portions of video that were actually used in the automated pose estimation, excluding the "... large fraction of frames where the tardigrades were either not in focus, in an undesired position, or obscured by algae." (lines 466-467).

We now cite previously published gait analyses and the number of strides, if they could be extracted, otherwise the duration. Many of these analyses only report animals walking in channels, and we now highlight that our animals can move within the 2D plane, potentially also allowing us to observe gaits adopted in turning, as we discuss based on our cluster analysis.

The authors further state that their extended data collection may enable 'the observation of gait transitions' (line 282). Such transitions have already been observed (in refs 51 and 52), and would not be a new avenue of analysis opened by this particular study. Discrete gait transitions are not readily observed in tardigrades; rather, they smoothly transition between coordination patterns (see ref 51).

We appreciate the reviewers' expert input on this topic. We have revised our discussion of the gait transitions and now refocus the relevance on long-term tracking and navigational goals.

Line 449 - change to "Brightfield tracking of tardigrades"

Done

Comments on FIGURES and FIGURE LEGENDS and TABLES

In the 1D legend, the top-to-bottom order of the magnification values in the images do not appear to correspond to the descriptions in the legend. Also, the 'magnification' values are hard to square with the scale bar lengths shown in the images. Consider just noting, for example, 3.125X rather than 3.125x magnification. Or (perhaps better) let the scale bars speak for themselves in these and in other images in the manuscript. Done

Figure 2 title: add the species and italicize that (i.e. rather than 'Larvae'). Done

Figure 2F the inset is a bit confusing because the lower track that veers off to the right appears to be from a later(?) time period than the rest of the inset. Consider omitting this lower track from the inset.

Done

Figure 2F legend: consider 'example' rather than 'exemplary'. Done

Figure 3C, D legend could be organized more logically (e.g. put the **** $p < 0.0001$ with D). It would be useful to add information in the legend (also for 3E) about what specifically is being compared in the statistical tests.

Done

Figure 3F legend: include more detail about the statistical test used. Done

Figure 4 legend would benefit from more stand-alone context and explanation regarding 'pumps' and (especially) 'prey signal'. Figure 4E states that 'prey signal' was detected as in C, but no information is given in the legend about how prey signal is detected in C, or what the term 'prey signal' means.

Done

Figure 4B legend is incomplete ("...predicted behavioral states using the model published in .")

Done

Figure 4D it may be useful to partition these 43 tracks according to the 8 predators, in order to compare variation between the predators.

We appreciate the suggestion but we prefer showing the heatmap sorted by prior state, as this illustrates from which states the animals transition to arrive in the biting state. We have used this machine-learning analysis and this tracking microscope in an unrelated manuscript studying aggression in predatory nematodes. The reference is now properly reproduced in the reference list - we apologize for the accidental omission previously.

Figure 5A is missing the 'A'. Done

Figure 5C it would be useful to include an outline (dotted white line?) around the entire *C. elegans* individual, for context. As before, the magnification value in 5C is a bit confusing in combination with the scale bar on the image (the image as shown appears to be magnified more than 3.125x). Consider just including the scale bar, or the specific objective used (EO 16 mm?). Done

Figure 5E is missing the 'E'. It would be useful to have Y-axis values shown here (e.g. animal 1, animal 2, etc.)
Done

Figure 5G - would be useful to have animal labels here too (as in 5C).
Done

Figure S1B - What is the significance (if any) of the outlier point at ~33Hz image acquisition?

We appreciate the reviewer pointing this out. Since the submission of the manuscript, our code has been substantially improved in runtime and how we parallelize the image acquisition, stage compensation, and auto-focusing threads. We therefore repeated this assessment and now find an intuitive relationship: If the acquisition rate is slow, we obtain a 1:1 ratio between frames acquired and tracking steps. If we increase the framerate of the camera to > 23 Hz, we automatically reduce the tracking steps to avoid calculating the tracking estimates on potentially blurry frames as the stage motion will not be completed in the non-exposing time window. Again, at even higher framerates, we see an even larger increase, where we start skipping two acquired frames per frame used for tracking. This is then also reflected in the tracking frequency compared to the acquisition frequency, which is non-monotonic.

Table S2: To put 'affordability' in real world context, it would be useful to add the total costs in Table S2, and report these values in the main text, and compare different configurations (brightfield, single color fluorescence, dual color fluorescence).

We have added the price for all the configurations to the website and the supplementary table S2.

Reviewer #1 (Remarks on code availability):

The Python code is well-commented, neat, and expertly constructed. A 'readme' file is available with installation and usage instructions. I did not download (or run) the code, as it primarily controls hardware I do not possess, and analysis of videos acquired from this hardware.

We appreciate the positive comments on our code and the suggestions for further improvement. We have addressed all technical comments where applicable, in the manuscript, documentation, or directly in the code. The technical comments and suggestions are very appreciated and make the documentation more useful for a broad user base.

Data is reported to be at osf.io/rpfny ... when I visited this site on 3 June 2025, it was not accessible - it said "You Need Permission".

We thank the reviewer for making us aware of this. The permissions have now been set correctly.

Reviewer #2 (Remarks to the Author):

Synopsis:

A modular multi-color fluorescence microscope for simultaneous tracking of cellular activity and behavior reports the development of a modular tracking microscope (or macroscope) that can incorporate neurobiological data using real-time recording of fluorescent indicators. This work fits an ongoing pursuit by a number of labs and follows publications such as Arous 2010 (10.1016/j.jneumeth.2010.01.011), Venkatachalam 2016 (10.1073/pnas.1507109113), Nguyen 2016 (10.1073/pnas.1507110112), Voleti 2019 (10.1038/s41592-019-0579-4), among others that the authors cite. The microscope reported here is cheaper than many of the others, and the authors have shown it is capable of tracking other small invertebrates like tardigrades and fly larvae (though this has also been reported before by Huang 2024 (10.1016/j.bbrc.2024.150290)). Though no novel capabilities are reported, the modularity, affordability, and open-source nature of the microscope and software make it a welcome addition to the field.

We thank the reviewer for their thoughtful assessment of our work and for recognizing the value of our modular, affordable, and open-source tracking microscope to the research community.

We appreciate the reviewer highlighting the context of our work within the broader field of tracking microscopy. We would like to emphasize that our system is specifically designed as a tracking microscope for imaging of cellular structures, organs, and behavioral features in freely moving animals, rather than for whole-brain imaging.

While the excellent whole-brain imaging systems cited by the reviewer (Nguyen 2016, Venkatachalam 2016, Voleti 2019) achieve pan-neuronal calcium imaging through sophisticated optical designs and higher magnifications, our microscope intentionally operates at lower magnifications (1-4.2x) with a larger field of view to enable long-term tracking of the entire unrestrained animals across centimeter-scale arenas. This design choice allows us to image specific neurons (as demonstrated with PLM touch receptor neurons in Figure 3), muscle activity patterns (Figure 2), and organ-level dynamics (Figure 5) while animals navigate complex environments over extended periods. We revised the manuscript to more explicitly state this distinction in the introduction and results sections.

Major manuscript comments:

Since a primary use-case is behavioral tracking, I think more could be included in the text about caveats associated with environmental control, lack thereof, or the potential for adding it, and how the imaging environment could impact behavior. For example, it isn't entirely obvious that the light source is filtered through a slit (only mentioned in figure captions), which I assume

means that the rest of the arena is in the dark (the Getting Started page of the documentation suggests this). The effects of a light gradient in the behavioral arena are difficult to predict, even if the tracked organism itself is always illuminated.

Can other types of environment control be added?

We sincerely thank the reviewer for raising the important point about the potential effects of illumination gradients on animal behavior. Ensuring controlled and uniform lighting is indeed crucial for behavioral experiments. To address this, we have now explicitly clarified in the manuscript that our system employs a slit aperture projected onto the sample to restrict excitation light to a narrowly defined, localized region with the same dimension as the camera field-of-view. This slit ensures that the tracked animal is consistently and evenly illuminated within this confined area, while the rest of the arena remains unilluminated. It is also required to avoid that the two spectrally-separated images do not overlap in the sensor. As the referee points out, we cannot exclude that light gradients at the rim of the illuminated area or as the illuminated area moves with the tracked animal might affect the behavior of other, non recorded animals. However, since the system actively tracks and keeps the animal centered in this illuminated area, the recorded animal does not experience variable lighting conditions that could induce gradients or bias its behavior.

Furthermore, our fixed-arena design facilitates the creation and maintenance of other environmental parameters commonly used in behavioral studies—such as chemical gradients (Figure 2), or social contexts (Figure 4)—without interference from illumination changes. We appreciate the reviewer’s comment, which helped improve the clarity of this aspect in the revised manuscript.

Lines 465-470 could use clarification. A subset of videos was used for training, but it’s unclear what subset. The next sentence reads as if the training set included best-case and edge-case (poor) frames, but then the next sentence seems to walk that back by suggesting only best-case frames were used for analysis. If edge-case frames were used for training, why were they ignored during prediction? What proportion of frames included a predicted model that surpassed a confidence threshold? Since a feature of the system is its potential for use with non-model organisms, it’s important to understand just how robust the imaging and tracking is in cases where perfection is not possible.

We appreciate the comments, and we have substantially revised the relevant methods section to explain and describe the procedure in detail.

What is the largest size of arena that could be potentially be used? Line 112 states that the motorized stage can move 150 mm – is that in both X and Y? Have the authors used an arena this large?

We thank the reviewer for this important question, which highlights the strength of this modular design. The motorized stage we used provides a travel range of approximately 110 mm (Y) × 80 mm (X). We have successfully performed behavioral tracking experiments on large 15 cm

diameter unseeded NGM plates, as demonstrated in the *Drosophila* larval chemotaxis assays presented in the manuscript (Figure 2).

In practice, the assembled microscope setup has an effective imaging area of about 100×140 mm² for typical experiments, which is smaller than the maximal stage limits due to partial occlusion by the stage hardware. These details have now been added to the 'Hardware' section of the revised manuscript to clarify the system's capabilities and limitations. We also added that stages of the same model with larger travel ranges can be used without affecting the setup or software.

Minor manuscript comments:

The authors need to decide if it's a microscope (manuscript) or a macroscope (documentation). Done, we decided on microscope.

Throughout it seems that "stage" is used to mean "camera," since it's the camera that is moving, not the stage that the plate is set upon. This needs fixing or clarifying.

We thank the reviewer for the helpful comment and the opportunity to clarify. The microscope compensates for animal motion by moving the camera and light source together on a motorized stage, while the sample remains fixed in place. This reduces vibration and allows integration of additional tools without disturbing the animal. We have defined "stage" as the motorized platform carrying the camera and light source - not the sample. We now consistently use this terminology in the revised text to ensure clarity.

Reviewer #2 (Remarks on code availability):

I reviewed the GitHub repo and documentation website but did not closely examine the code for tracking implementation.

More should be said about the data that comes out of GlowTracker, including file formats, compression, typical file sizes, etc.

We have added a description and a table to the supplementary information detailing the output files and image formats generated by the software. In addition, the same information is now also available on the documentation website.

The tardigrade videos were seemingly imported to DeepLabCut – is cross-platform integration of the output files straightforward? Do users get access to both videos (i.e., MP4 or AVI) and coordinate data (CSV, HDF5, other)?

We have added descriptions of the output formats as well as the coordinate files generated during tracking to the manuscript and online documentation. The resulting stack of tif files can be used in DeepLabCut by converting to .avi files. We have added detailed instructions to the methods section.

I cannot find documentation on initiating tracking of a specific animal. The tracking_explanation page includes a lot of technical information that is interesting but superfluous for most users. In a multi-animal experiment, such as the examples on that page or the predator vs. prey example in the manuscript, how does the user select the animal to track? Is tracking fidelity maintained upon animal collisions/overlap? This is an important question to answer about the experiment highlighted in Figure 4.

We appreciate this comment and have expanded the supplementary information, as well as added an instructional video to the website. We thank the reviewer for pointing out this oversight.

In Figure 4, we can seamlessly retain the identity of the animal as the predator and prey appear in different colors (colors are projected side-by-side onto the camera chip). Users can pre-select if they wish to track, e.g., GFP or mCherry labeled animals, and only that color will be used to track the animals.

I was able to install and start the GUI on a Mac Studio (Apple Silicon), though unable to test the functionality without the camera. I will note that mixing conda environments and pip installs is not best-practice. I highly recommend uv for package management, or at least synchronicity with pip and venv or virtualenv. For potentially broader adoption by non-technical users, the software could be containerized for launch with Docker Desktop bundled with a launcher like PyInstaller.

Linux and Windows are mentioned in the documentation, but I can't find anywhere that lists OS's that have been confirmed to work.

We appreciate the suggestion and have added the specific OS we have tested to Supplementary Table 3. We also perform automated installation tests for three major OS upon code updates using GitHub actions, ensuring that the package remains compatible with the latest OS releases. We have additionally unified the installation instructions to use 'venv'. We appreciate the suggestion for dockerization, which we will take into account in future releases.

Reviewer #3 (Remarks to the Author):

The objective of the paper was to build a simple microscopy tool consisting features like live tracking of small model organisms (C elegans, fly larva etc etc) with fluorescence set up (single and dual colour) as well as bright field microscopy (tardigrade). They also combined some quantitative measurement of neural activity with genetically encoded calcium sensor (GCaMP) within the neuron of transparent models. They also suggest that the system is easy to assemble by non-experts. They have given some experimental examples in the nematode C. elegans, Drosophila larvae and tardigrade.

According to me it is an interesting paper discussing a microscopy tool development, which is helpful for researcher in general. However, I would like to point out that there are a number of papers which deals with similar methodology including few in worm. Every microscopy

techniques and assays they have used are already established before and they have provided reference too. I am not sure whether this is appropriate for Nature Communication as I don't see any new findings as well as these tools are already used by researchers in various models. I agree that these methodology are useful, but the authors at least could not convince how these microscopy-tools would solve any question that is unanswered as of now.

We thank the reviewer for their thoughtful assessment and for recognizing the value of our modular tracking microscope to the research community. We apologize for not clearly enunciating the novelty of our approach in the prior manuscript and prioritizing technical findings over biological context. In the revised text, we made many changes to better reflect which aspects are novel. However, due to space limitations and to give sufficient technical details for the users, as requested by the other reviewers, we have not included all of the suggested additional measurements. We believe these would be better placed in a dedicated paper discussing these topics in much more depth.

We would argue that the innovation in our manuscript lies in providing fully open-source, affordable systems that can be built and used without specialized engineering or coding expertise, filling a key gap in reproducibility and accessibility. While similar systems exist, these are barely documented, substantially more expensive (>200k USD compared to our <20k), and assembly is never described. Requiring substantial new biological discoveries would, in our view, shift the focus away from these central objectives and detract from the paper's utility as a resource. Instead, we focus on rigorous validation of system performance, particularly that low-cost components can match the capabilities of >\$200,000 commercial microscopes typically used for these experiments. One user-assembled system achieves what typically requires separate equipment: single-neuron calcium imaging (*C. elegans* touch receptors), muscle dynamics (*Drosophila* larvae), gait analysis in non-model organisms (tardigrades), and dual-color predator-prey tracking. This versatility means laboratories can pursue diverse research questions with a single investment rather than choosing a single experimental focus based on equipment limitations.

We also want to highlight that this study shows the dual-color visualization of predator-prey interactions between *P. pacificus* and *C. elegans* in freely moving animals. Our system maintains clear animal identity throughout contact events—predator (pharyngeal RFP) and prey (body wall GFP) appear in distinct colors. Importantly, animal contact becomes visible as prey GFP signal at the predator's pharynx, allowing us to distinguish actual biting and feeding from exploratory behavior. Previous *P. pacificus* studies relied on behavioral inference or counting dead larvae afterward. Now, researchers can observe exactly when and how predation occurs, validating behavioral models with direct biological evidence.

We have now revised the text to clearly describe the novelty (technical and biological).

I am also listing few specific questions below in regard to the presented data in this manuscript.

1) This paper majorly focuses on cost-effective and vibration free tracking system for Sample with Lightsource moving while tracking. Their epifluorescence system cost around ~14000

Euros where else similar dual epifluorescence vibration free system WormsPY (2024) cost around ~12300 Euros with easily available parts. See the reference # 53.

We thank the reviewer for mentioning WormsPy. We had co-submitted this manuscript with our colleagues who developed Wormsy, as there are some similarities, however, unfortunately they seemed to have undergone a separation in the reviewing process. Below, we detail the differences between our setups and the complementary experiments enabled with these tools. We summarize these details briefly in our revised discussion.

While both systems offer vibration-free tracking with moving optics, they serve fundamentally different experimental needs. WormsPy operates at high magnification (7.5x) within a restricted field of view (1.4 × 0.9 mm), making it excellent for detailed neural calcium imaging optimized for single organisms in small arenas. In contrast, our system operates at lower magnifications (1-4.2x) with a much larger field of view (up to 7.4 × 5.0 mm), enabling tracking across 15 cm plates, multi-species experiments (*C. elegans*, *Drosophila* larvae, tardigrades), and critically, dual-color predator-prey interactions that would be infeasible with WormsPy's limited FOV.

Regarding cost, while Wormsy costs ~€12,300 for a specialized *C. elegans* imaging system, our €14,500 system includes components for three modular configurations (brightfield/single-color/dual-color) and larger motorized stages (150 mm travel range). This slightly higher investment provides access to diverse experimental paradigms—from single-neuron calcium imaging to ecological predator-prey studies—making advanced imaging accessible to laboratories studying various organisms and behaviors. Essentially, our system serves as a versatile platform for broader biological investigations across species and scales.

2) Figure-1: You have shown some brightfield images of tardigrade and larvae in upcoming results in the same microscope. Can you put light path of brightfield also in Figure 1 ? like you have done red and green.

We appreciate the suggestion, but as putting all 3 modular configurations would take a large amount of space, we have kept these lightpaths in the Supplementary figure.

Suggestion: In figure 1D Can you show GCaMP and mCherry in separate channel like you did in Fig 3B? It is easier to see

OK

3) Figure 2: In this result section authors describe, how fly larvae crawls towards an attractive odours. They tracked the larvae with fluorescently labelled neuron. In figure-D-E: They show how the Odors experienced by the successful (blue) and unsuccessful (red) larvae over time. Not sure why one has to use fluorescent larvae for this as simple brightfield is sufficient to capture and track the movements.

Also, it's not clear from this the relationship between neural activity and successful vs unsuccessful trial. Since GCaMP sensor is present in the sensory neuron, one expects some neural activity could be monitored in this case.

We appreciate the reviewer's comment and apologize for any confusion. We would like to clarify that in Figure 2, we are imaging **muscle activity**. The larvae express GCaMP6m in body wall muscles via MEF2-Gal4 driver (not in sensory neurons), allowing us to visualize peristaltic muscle contractions during chemotaxis.

While the reviewer correctly notes that brightfield imaging alone could track larval trajectories, our dual-color approach simultaneously captures both behavior and underlying muscle dynamics. Figure 2G demonstrates this capability by showing kymographs of calcium waves propagating along the anterior-posterior axis during locomotion—physiological data that would be impossible to obtain with brightfield alone. These muscle contraction patterns are fundamental to understanding the biomechanics of larval crawling (Heckscher et al., 2012, *J. Neurosci.*) and demonstrate our system's capability to extract physiological information across different scales while maintaining behavioral tracking.

1F-G: Does there is GCaMP6 activity differ in muscle of motile larvae vs non-motile larvae? That comparison kymographs will be interesting to see.

As GCaMP reports on muscle contractions, we do not expect to see transients in the muscles of non-motile larvae.

4) Figure #3: It's not clear where the piezo is kept on NGM plate. Please have a clear illustration.

Done

I would also like to see the GCaMP recording from ALM neuron as well, which might be responsible for the reversal behaviour in response to mechanical cue.

I would also like to see more details of the behavioural quantification in response to mechanical stimuli. What percentage of worms show reversal vs forward movement?

How can one rule out the effect of other mechanosensory neurons such as PVD in this experiment?

We appreciate the reviewers interest in the mechanical stimulation experiments. The gentle touch circuit in *C. elegans* is very well characterized and has been shown to respond well to 630 Hz buzz as we apply here (<https://pubmed.ncbi.nlm.nih.gov/31533952/>). The indentation strength of the piezo buzzer does not trigger the noxious harsh touch system of the animals, as *mec-4* mutants do not display an escape response:

(data not recorded on this microscope and used in another manuscript). We have also used this stimulus presentation in a detailed characterization of the PLM neurons:

<https://pmc.ncbi.nlm.nih.gov/articles/PMC11457936/>.

We have expanded our description of these experiments and cited more relevant literature describing the neuronal circuit involved, which has been very well characterized.

The authors should test the behavioural response in both *mec-3* (null for large force) and *mec-4* (null for gentle force) mutants.

We appreciate the reviewers' interest in this question. Since these experiments are well established, and we and others have previously published these data using conventional calcium imaging microscopes, we do not believe this substantially adds to this manuscript.

5) Figure 4: Visualising animal and environment interaction

Query: Can you show the brightfield video frames of prey-predator interaction. With just fluorescence imaging, it is not clear

We appreciate the suggestion and have redesigned the figure to include a brightfield image of the predator prey interaction.

6) Figure 5: Modular single-color design enables long-term tracking.

Query: Is there any condition where pharyngeal pumping significantly changes? It can validate the tool further for long-term assay

For example *itr-1* mutant (lacking IP3 signalling).

We thank the reviewer for this suggestion. While testing specific mutants like *itr-1* is beyond the scope of this method paper, we agree that pharyngeal pumping is well-established as a sensitive readout of physiological state that changes under various conditions. The literature, including work done in our group, extensively documents pharyngeal pumping modulation by multiple factors: serotonin increases pumping rates (Song & Avery, 2012, Lee et al., 2027), food deprivation alters pumping (E. Bonnard et al. 2022), prior history and life history affect it, too (Song and Avery, 2025).

Our demonstration of stable pharyngeal GCaMP8f imaging over 1 hour (Figure 5D-E) with minimal photobleaching establishes that our system can extend typical imaging times and makes our platform suitable for studying long-term changes in feeding behavior, internal states, or learning, processes that are not captured in the more typical minute-long assessments.

7) Figure 6: Bright-field tracking for non-model species

Query: Only 9 videos were used to train the model, How many tardigrades were there in each video?

We track individual animals, and as such there was one animal per video. However, we have acquired an entirely new dataset for this revision, and included more samples, longer videos and an overall improved quantitative analysis of these data.

8) Also, it would be nice if they can comment and give details whether their system would allow optogenetic manipulation (Channelrhodopsins) of neurons and simultaneous tracking of behaviour or imaging of GCaMP response in neurons. This is another important component needs to be clear.

We appreciate this suggestion and have added a paragraph to the discussion describing this interesting extension!

Reviewer #3 (Remarks on code availability):

The authors did a good job in detailing the methods and codes. The code provide the README file and instructions are there for installation.

I was able to install and run the code.

We appreciate the reviewers positive comment on code documentation and usability.

Response to REVIEWERS' COMMENTS

Reviewer #1 (Remarks to the Author):

The authors develop a low-cost accessible system, including both hardware and software, for tracking and quantification movement, at impressive scales in both space and time. They demonstrate the performance of their hardware and software by tracking a variety of species, including whole-animal tracking via fluorescent markers, real-time calcium imaging, interactions between species, and extendability to multiple kinds of moving animals. The inclusion of a detailed parts list with prices is especially invaluable, as it will allow readers to compare with alternative systems and to modify the parts to suit their needs. In all, the authors describe a set of tools that will be of interest to researchers looking to quantify locomotor behaviors in real time. I appreciate the honest and thorough efforts the authors have made to address reviewer concerns, and I especially appreciate the completely new analyses of tardigrade stepping patterns. I still have a few misgivings about the accuracy of the automated step tracking, and the determinations of stances vs. swing phases, but the inclusion of the anterior-posterior positioning of each leg allows the reader to evaluate the accuracy of the phase determination on their own. The authors also compare the accuracy of automated tracking with manual tracking. These are minor points, however, as the paper is more about hardware and software development than it is about the intricacies of tracking leg motion in a particular species.

Some suggestions that would further strengthen this section of the paper:

1. As an additional measure of accuracy of automated stance-swing annotations, I suggest that the authors compare the duty factors obtained in their dataset (ideally in a plot vs. speed) with those of previous studies on tardigrades (refs 53 and 54). This comparison would not require additional experiments - just extraction of additional parameters from their existing data. This analysis would help determine if there are systematic errors in their phase-calling (e.g. if swing phases are too long).

We thank the reviewer for their valuable comments. We calculated the duty factor, and could show that this falls in the range of previously reported values (ref 54 and 55). The correlation between duty factor and speed is similar to what was reported by (ref 53). The analysis was added as Supplementary Figure 5G.

2. Also, I would suggest reversing the colors so that swing phases are shown in black, which seems to be the convention (see for example Mendes et al. 2013, DOI: 10.7554/eLife.00231, Wosnitza et al. 2013, doi:10.1242/jeb.078139, and others).

Done

3. Include a legend with the colors for swings and stances for Figure 6F and 6G.

Done

4. For Figure 6F, I would also recommend reorganizing the legs according to their side (first) and then segment, as done in the aforementioned studies in *Drosophila*, and the tardigrade walking studies cited (refs 53 and 54).

Done

Finally, double check citations throughout the paper. For example line 334: “The normalized mean anterior-posterior position of the different cycles show similarities with gaits previously described in tardigrades (Fig. 6G) (56, 57)” ... but 56 and 57 are studies in *Drosophila*.

Done

Reviewer #1 (Remarks on code availability):

Code is well-documented. The README file provides easily-understood instructions for installation and use.

Reviewer #2 (Remarks to the Author):

I thank the authors for their thoughtful responses to my comments.

Reviewer #2 (Remarks on code availability):

The README and documentation are sufficient. I was able to install and run the code.

Reviewer #3 (Remarks to the Author):

The authors have addressed all the questions and the revised manuscript is suitable for publication.

Reviewer #3 (Remarks on code availability):

The authors did a good job in detailing the methods and codes. The code provide the README file and instructions are there for installation. I was able to install and run the code.